# Focus On What Matters: Separated Models For Visual-Based RL Generalization

**Di Zhang    Bowen Lv    Hai Zhang    Feifan Yang    Junqiao Zhao** [*]
**Hang Yu    Chang Huang    Hongtu Zhou    Chen Ye    Changjun Jiang**
Department of Computer Science, Tongji University, Shanghai, China
MOE Key Lab of Embedded System and Service Computing, Tongji University, Shanghai, China
`{2331922, 2151769, zhanghai12138, 2153299, zhaojunqiao}@tongji.edu.cn`
`{2053881, 2130790, zhouhongtu, yechen, cjjiang}@tongji.edu.cn`

## Abstract

A primary challenge for visual-based Reinforcement Learning (RL) is to generalize effectively across unseen environments. Although previous studies have explored different auxiliary tasks to enhance generalization, few adopt image reconstruction due to concerns about exacerbating overfitting to task-irrelevant features during training. Perceiving the pre-eminence of image reconstruction in representation learning, we propose SMG (Separated Models for Generalization), a novel approach that exploits image reconstruction for generalization. SMG introduces two model branches to extract task-relevant and task-irrelevant representations separately from visual observations via cooperatively reconstruction. Built upon this architecture, we further emphasize the importance of task-relevant features for generalization. Specifically, SMG incorporates two additional consistency losses to guide the agent's focus toward task-relevant areas across different scenarios, thereby achieving free from overfitting. Extensive experiments in DMC demonstrate the SOTA performance of SMG in generalization, particularly excelling in video-background settings. Evaluations on robotic manipulation tasks further confirm the robustness of SMG in real-world applications. Source code is available at `https://anonymous.4open.science/r/SMG/`.

## 1 Introduction

Visual-based Reinforcement Learning (RL) has demonstrated remarkable success across various tasks, including Atari games [27, 11, 18], robotic manipulation [23, 9], and autonomous navigation [26, 46]. However, deploying visual-based RL algorithms in real-world applications requires a high generalization ability due to numerous factors that can induce distribution shifts between training and deployment scenarios, such as variations in lighting conditions, camera viewpoints, and backgrounds. Many visual-based RL algorithms are prone to overfitting to the training observations [5, 34, 44], limiting their applicability in scenarios where fine-tuning with deployment observations is not allowed.

To address the generalization gap in visual-based RL, current studies primarily focus on utilizing data augmentation techniques [19, 20, 33] and exploring various auxiliary tasks [12, 3, 13]. However, few of the previous works successfully incorporate reconstruction loss to this field, which is commonly adopted in standard visual-based RL settings and has been demonstrated to improve the sample efficiency of RL agents [41, 10, 7]. This is because reconstructing the entire input observation can exacerbate the overfitting problem to task-irrelevant features and thus weaken the generalization ability. Although several works also explored extracting task-relevant features from visual observations

---

[*]Corresponding author

38th Conference on Neural Information Processing Systems (NeurIPS 2024).

[6, 39, 45], little attention has been paid to the potential of leveraging these features in improving generalization.

In this paper, we propose SMG (Separated Models for Generalization), a method that utilizes a reconstruction-based auxiliary task to extract task-relevant representations from visual observations and further strengthens the generalization ability of RL agents with the help of two consistency losses. The core mechanisms behind SMG can be summarized in two parts: First, we introduce two model branches to disentangle foreground and background representations underlying in the visual observations. This separated model framework circumvents the risk of overfitting task-irrelevant features inherent in a single model structure by prudently designing the reconstruction paths, allowing our model to benefit from reconstruction loss without sacrificing generalization ability. Second, we introduce two consistency losses to align the agent's focus on the task-relevant features between raw and augmented observations. This approach enables the foreground model to extract more robust task-relevant representations, which substantially boost the generalization capability of RL agents across diverse deployment scenarios.

We evaluate SMG's effectiveness across a range of challenging visual-based RL tasks, including five tasks from DMControl [36] and two more realistic robotic manipulation tasks [17]. We also adapt different evaluation settings with random-color and video-background modifications. Through comparisons with strong baseline methods, SMG demonstrates state-of-the-art performance in terms of generalization, particularly showcasing superiority in video-background settings and robotic manipulation tasks.

In summary, the main contributions of this paper are as follows:

- We present SMG, a novel approach that aims to enhance the zero-shot generalization ability of RL agents. SMG is designed as a plug-and-play method that seamlessly integrates with existing standard off-policy RL algorithms.

- SMG emphasizes the significance of task-relevant features in visual-based RL generalization and successfully incorporates a reconstruction loss into this setting.

- Extensive experimental results demonstrate that SMG achieves state-of-the-art performance across various visual-based RL tasks, particularly excelling in video-background settings and robotic manipulation tasks.

## 2 Background

A Markov Decision Process (MDP) can be defined as a tuple $(\mathcal{S}, \mathcal{A}, p, r, \gamma)$, where $\mathcal{S}$ is the state space, $\mathcal{A}$ is the action space, $p : \mathcal{S} \times \mathcal{A} \times \mathcal{S} \rightarrow [0, 1]$ is the state transition probability function, $r : \mathcal{S} \times \mathcal{A} \times \mathcal{S} \rightarrow \mathbb{R}$ is the reward function, and $\gamma \in [0, 1]$ is the discount factor. At each time step $t$, the agent receives a state $s_t \in \mathcal{S}$, selects an action $a_t \in \mathcal{A}$, and then receives a reward $r_t \in \mathbb{R}$. The agent's goal is to learn a optimal policy $\pi(a_t|s_t)$ that maximizes the expected return $\mathbb{E}_{(s_t,a_t)\sim\rho_\pi}[\sum_{t=0}^{\infty} \gamma^t r_t]$, where $\rho_\pi$ defines the discounted state-action visitation of $\pi$.

Learning an optimal policy from visual observations poses a substantial challenge for RL agents due to the inherent partial observability of the environment, a characteristic of POMDPs (Partially Observed MDP). For one thing, at each timestep $t$, the visual observation $o_t$ can only capture partial information about the true state $s_t$, as certain elements may be obscured in the image. For another, the dimension of $o_t$ is much higher than that of $s_t$, which makes it difficult to utilize $o_t$ directly for policy learning.

To infer the true underlying state from visual observations, existing methods usually employ a parameterized encoder $f$ to map a stacked frame sequence $x_t = (o_{t'}, o_{t'+1}, ..., o_t)$ to a compact low-dimensional latent vector $z_t$, which is then used as input by policy and value function. However, training the encoder solely to rely on the reward signal is demonstrated to sample inefficiency and may lead to suboptimal performance [41]. To tackle this issue, various auxiliary tasks have been proposed to enhance encoder training, with one common choice being to extract features from pixels via image reconstruction loss [7, 21, 2]. By adding another parameterized image decoder $g$, the reconstruction loss is defined by maximizing the likelihood function:

$$L_{\text{recon}} = -\mathbb{E}_{o_t \sim \mathcal{D}}[\mathbb{E}_{z_t \sim f(o_t)}[\log g(o_t|z_t)]] \tag{1}$$

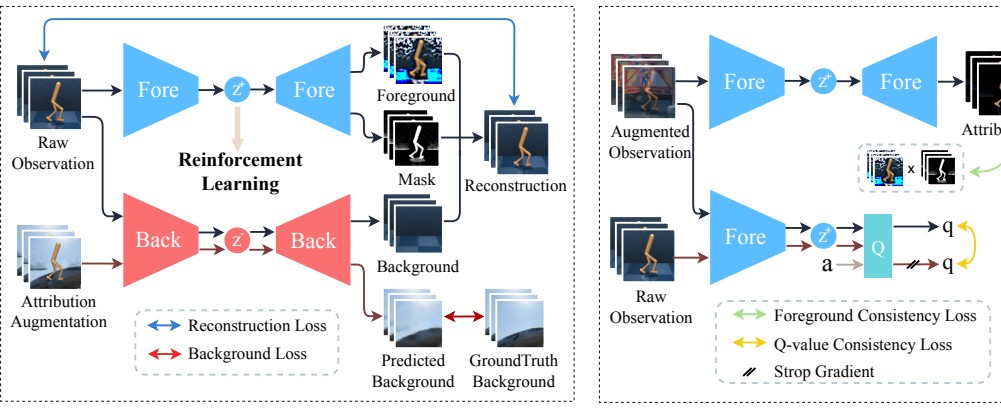

(a) Learning Task-Relevant Representations With SMG      (b) Improving GeneralizationWith SMG

Figure 1: Architecture of SMG. One-way arrows represent different types of data flows with the same input. Two-way arrows represent different types of loss.

# 3 Approach

## 3.1 What Matters in a Reinforcement Learning Task?

Learning to generalize is hard for RL agents, particularly when utilizing an image reconstruction loss. While images are rich in information, requiring the agent to reconstruct the entire input observation can lead the autoencoder network to overfit to features that are unrelated to the task (e.g. colors, textures, and backgrounds). In contrast, humans can accurately figure out what matters visually when learning a new task. Even when colors or backgrounds are changed, humans can still leverage the prior knowledge to complete the task by focusing on task-relevant

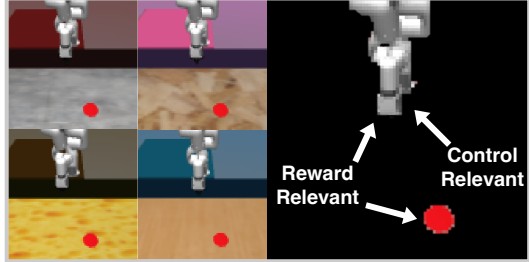

Figure 2: A robotic manipulation task explanation for task-relevant parts in the environment.

features. Considering a robotic manipulation task where the agent must move the arm to the red target (Figure 2), despite variations in background colors and textures across four test scenarios on the left, only the arm's orientation and the target position should be focused on this task. We aim for our RL agent to learn an optimal policy that solely relies on these task-relevant features while disregarding irrelevant regions.

Formally, we decompose the latent representation $z_t$ into task-relevant part $z_t^+$ and task-irrelevant part $z_t^-$. These two representations are independent, as $p(z_t|o_t) = p(z_t^+|o_t)p(z_t^-|o_t)$. The task-relevant representation can be further subdivided into the "control-relevant" part, which is directly affected by the agent's actions (the arm); and the "reward-relevant" part, which is associated with the reward signal (the arm and the target), both are crucial for policy learning.

## 3.2 Learning Task-Relevant Representations with Separated Models

### 3.2.1 Separated Models and Reconstruction

The representation learning objective of SMG is to maximize the mutual information $I(o_t; z_t)$ between the observation $o_t$ and the latent representation $z_t$, and we further derive an image reconstruction objective incorporating the combination of task-relevant representation $z_t^+$ and task-irrelevant representation $z_t^-$ as follows:

$$L_{\text{recon}} = -I(o_t; z_t) \leq -\mathbb{E}_{o_t \sim \mathcal{D}}[\mathbb{E}_{z_t^+ \sim f^+(o_t), z_t^- \sim f^-(o_t)}[\log q(o_t|z_t^+, z_t^-)]] \quad (2)$$

Inspired by previous works [6, 30] that explore how to mitigate background distractions, we implement the reconstruction process by introducing the foreground encoder $f^+$ and the background

encoder $f^-$ to extract different types of representations simultaneously, which forms a separated models architecture. We also incorporate two decoders. The foreground decoder $g^+$ is employed to reconstruct the foreground image $o_t^+$ and predict a mask $M_t$ with values between $(0, 1)$. The background decoder $g^-$ is employed to reconstruct the background image $o_t^-$. The full image $o_t$ is then reconstructed by $o_t^+$, $o_t^-$ and the mask $M_t$ via $o_t' = o_t^+ \odot M_t + o_t^- \odot (1 - M_t)$ ($\odot$ denotes the Hadamard product), the reconstruction process is illustrated by the black arrows in Figure 1a. Notably, the area where the agent is focusing can be visualized as $o_t^+ \odot M_t$, which we term the "attribution" of the agent, formally defined as $Attrib(o_t)$.

### 3.2.2 Additional Loss Terms

Based on the separated models architecture, we define four additional loss terms to enhance the model's ability to distinguish between two types of representations. These include the mask ratio loss and background reconstruction loss, which supervise the model's pixel outputs; along with the Q-value loss and empowerment loss, designed to consider the two properties of task-relevant representation.

**Mask ratio loss.** To further refine the accuracy of mask prediction, we introduce a hyperparameter $\rho$, termed the mask ratio, to constrain the proportion of the foreground part in the mask. As shown in Equation 3, we regard $L_{\text{mask}}$ as an explicit form of an information bottleneck, as the percentage $\rho$ determines the number of pixels of $o_t^+$ retained in the final reconstruction. This constraint forces $f^+$ to prioritize the task-relevant parts of the observation during encoding. Empirical results in Section 4.4 demonstrate that $L_{\text{mask}}$ facilitates learning a more precise mask.

$$L_{mask} = \left( \frac{\sum_{i,j} M_t(i,j)}{\text{image\_size}^2} - \rho \right)^2 \tag{3}$$

**Background reconstruction loss.** Improving the precision of background prediction can consequently enhance the foreground as well. Since the foreground and background are complementary, providing supervision for the background prevents the foreground from learning all parts of the observation. Therefore, we add additional supervision to the task-irrelevant representation $z_t^-$. To achieve this, we propose a new type of data augmentation called attribution augmentation tailored for SMG, as illustrated in Figure 3b. This augmentation involves augmenting the raw observation $o_t$ with its corresponding predicted mask $M_t$ via $\tau_{\text{attrib}}(o_t) = o_t \odot M_t + \epsilon \odot (1 - M_t)$, where $\epsilon$ represents a randomly sampled image. This simulates the video-background setting in deployment scenarios. We define the background reconstruction loss $L_{\text{back}}$ as follows:

$$L_{\text{back}} = -\mathbb{E}_{o_t \sim \mathcal{D}} [\mathbb{E}_{z_t^- \sim f^- (\tau_{\text{attrib}}(o_t))} [\log g^- (\epsilon | z_t^-)]] \tag{4}$$

**Q-value loss.** Recall that the task-relevant representation $z_t^+$ has two key properties: reward-relevant and control-relevant. Satisfying the former is relatively straightforward, as the representation $z_t^+$ is used for policy learning. Through the Bellman residual update objective [35] outlined in Equation 5, $z_t^+$ will progressively enhance its correlation with the reward signal.

$$L_{\text{q}} = \mathbb{E}_{\tau \sim \mathcal{D}} [(Q(z_t^+, a_t) - (r_t + \gamma V(z_{t+1}^+)))^2] \tag{5}$$

**Empowerment loss.** For the control-relevant property, we integrate an empowerment term $I(a_t, z_{t+1}^+ | z_t^+)$ [28] based on conditional mutual information, which quantifies the relevance between the action and latent representation. Maximizing the empowerment term further leads to maximizing a variational lower bound $q(a_t | z_{t+1}^+, z_t^+)$ as shown in Equation 6. This objective necessitates that $a_t$ is predictable when two neighboring representations are known. We implement this objective by incorporating an inverse dynamic model.

$$L_{\text{action}} = -I(a_t, z_{t+1}^+ | z_t^+) \leq -\mathbb{E}_{p(a_t, z_{t+1}^+, z_t^+)} [\log q(a_t | z_{t+1}^+, z_t^+)] \tag{6}$$

The whole separated models architecture is shown in figure 1a.

### 3.3 Generalize Task-Relevant Representations with Separated Models

Utilizing the separated models architecture, SMG can successfully extract task-relevant representations from raw observations. Nevertheless, the agent still lacks the ability to generalize effectively

and may struggle to extract meaningful features from scenarios with transformed styles. To address this issue, we treat the task-relevant representation under raw observations as the ground truth and train SMG on more diversely augmented samples. Instead of directly optimizing the distance between the representations under raw and augmented observations, we introduce two types of consistency losses, considering both attribution and Q-values for more explainable supervision. By doing so, the foreground model can learn to extract task-relevant representations across different deployment scenarios.

**Foreground consistency loss.** To force the agent to focus on the same task-relevant area in transformed scenarios, we train the foreground models to predict the attribution under augmented observation $Attrib(\tau(o_t))$ with the supervision of the ground truth attribution $Attrib(o_t)$ (as $Attrib(o_t)$ is relatively easier to converge to an accurate value, and we discuss it in detail in Appendix F). The foreground consistency loss $L_{\text{fore\_consist}}$ is defined as Equation 7 (where **sg** means the stop-gradient operation).

$$L_{\text{fore\_consist}} = \mathbb{E}_{o_t \sim \mathcal{D}}[||Attrib(\tau(o_t)) - \mathbf{sg}(Attrib(o_t))||] \tag{7}$$

**Q-value consistency loss.** In addition to the attributions, the Q-values obtained from transformed observations also exhibit high variance [14], indicating instability in both the extracted representations and the Q function. To address this, we regularize the Q-values under augmented observations to be consistent with those under raw observations, as shown in Equation 8. This approach also regularizes the agent to learn an accurate task-relevant representation, as the gradient of $L_{q\_consist}$ is back-propagated to the latent space.

$$L_{\text{q\_consist}} = \mathbb{E}_{o_t,a_t \sim \mathcal{D}}[[Q(f^+(\tau(o_t)), a_t) - \mathbf{sg}(Q(f^+(o_t), a_t))]^2] \tag{8}$$

The above two consistency losses are illustrated in Figure 1b.

### 3.4 Overall Objective

Our proposed separated models architecture can seamlessly integrate as a plug-and-play module into any existing off-policy RL algorithms. In this work, we leverage SAC [8] as the base algorithm. Throughout the training phase, SMG iteratively performs exploration, critic update, policy update, and auxiliary task update. We define the critic loss $L_{\text{critic}}$ as the sum of the Q-value loss $L_q$ and the Q-value consistency loss $L_{\text{q\_consist}}$:

$$L_{\text{critic}} = L_q + \lambda_{\text{q\_consist}} L_{\text{q\_consist}} \tag{9}$$

Additionally, the auxiliary loss $L_{\text{aux}}$ comprises five previously mentioned loss terms:

$$L_{\text{aux}} = \lambda_{\text{recon}} L_{\text{recon}} + \lambda_{\text{mask}} L_{\text{mask}} + \lambda_{\text{back}} L_{\text{back}} + \lambda_{\text{action}} L_{\text{action}} + \lambda_{\text{fore\_consist}} L_{\text{fore\_consist}} \tag{10}$$

Although $L_{\text{aux}}$ contains five loss terms, experimental results show that using average weights for the first four terms and a smaller weight for the last term can achieve satisfactory performance. Detailed information about hyperparameters tuning is provided in Appendix C.3. The detailed derivation of Equation 2 and Equation 6 are provided in Appendix A.

## 4 Experimental Results

### 4.1 Setup

We benchmark SMG against the following baselines: (1) SAC [8], serving as the foundational algorithm for all other baselines; (2) DrQ [19], utilizing random shift augmentation; (3) SODA [12], incorporating a consistency loss on latent representations; (4) SVEA [14], focusing on stabilizing Q-values; (5) SRM [15], proposing a novel data augmentation technique; (6) SGQN [3], the previous SOTA method integrating saliency maps into RL tasks. We reproduce the results using the same settings reported in the original papers, with the exception of setting the batch size to 64 for all methods. Additionally, all results are calculated by four random seeds.

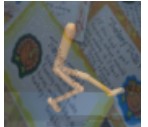 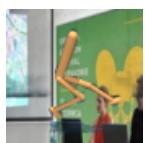

(a) Overlay      (b) Attribution

Figure 3: Two types of data augmentations using in SMG.

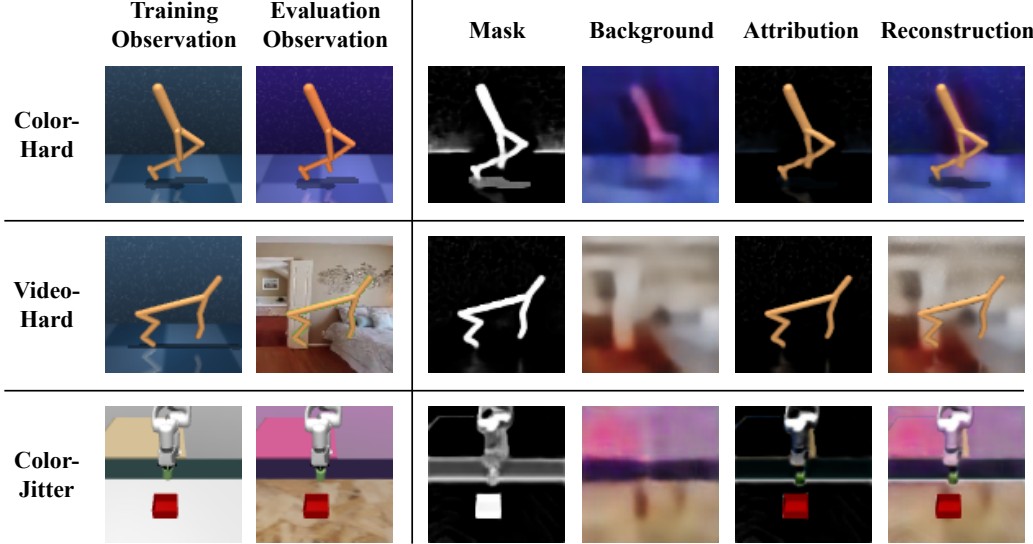

Figure 4: Visualizing the reconstruction process of SMG in different tasks (from top to bottom: *walker-walk*, *cheetah-run*, *peg in box*).

To achieve stable performance across various evaluation settings, we train SMG using a hybrid data augmentation approach for $\tau(o_t)$, involving random overlay [14] and attribution augmentation for all tasks (each time we randomly select a type of data augmentation, as shown in Figure 3). The network design for SMG and more detailed experiment settings are reported in Appendix C.

## 4.2 DMControl Results

We first conduct experiments on five selected tasks from DMControl [36] and adopt the same evaluation setting as DMControl Generalization Benchmark [12] (DMC-GB) used, which contains random-colors and video-background modifications across four different levels: *color-easy*, *color-hard*, *video-easy* and *video-hard*. Figure 5 shows an example in *walker-walk* task. We train all methods for 500k steps (except *walker-stand* for 250k, as it converges faster) on the training setting and evaluate the zero-shot generalization performance on the four evaluation settings.



|     (a) *Training*     |     (b) *Color-easy*     |     (c) *Color-hard*     |     (d) *Video-easy*     |     (e) *Video-hard*     |

Figure 5: Example of training and testing observation for DMC-GB (*walker-walk*). (a) is the training observation. (b-c) indicates different degrees of color change; (d-e) replaces the background with random videos, with (e) additionally removing the floor and the walker's shadow.

To provide a clear explanation of how SMG reconstructs images, we present the image outputs of *walker-walk* and *cheetah-run* after 500k training steps of training in the first two rows of Figure 4. The last four columns illustrate the model outputs necessary for reconstructing the evaluation observations. The predicted attribution (the fifth column) highlights the extracted task-relevant area, which shows SMG accurately depicts the attribution of the input observation while omitting the task-irrelevant elements such as the skybox, the floor, and even the random color variation. This indicates that the task-relevant representation $z_t^+$ contains only the information required to accomplish the task, which is crucial for generalization. Note that we aim to maintain the similarity between $Attrib(\tau(o_t))$ and $Attrib(o_t)$, even in random-color settings. As shown by the first row of *color-hard* setting, SMG predicts a yellow attribution despite the input evaluation observation being orange.

Table 1: DMControl results in video-background settings. We evaluate each seed five times and calculate the mean value. Then, we calculate the mean and standard deviation with four random seeds. Red indicates the best and blue indicates the second-best. Δ = improvement of SMG over the second best.

| DMControl (video-easy) | SAC | DrQ | SODA | SVEA (overlay) | SRM | SGQN | SMG (ours) | Δ |
|---|---|---|---|---|---|---|---|---|
| cartpole, swingup | 175 ±23 | 606 ±31 | 617 ±76 | **718** ±101 | 645 ±108 | 717 ±77 | **839** ±16 | +121 17% |
| finger, spin | 171 ±37 | 511 ±192 | 615 ±56 | 817 ±94 | 642 ±101 | **860** ±82 | **952** ±48 | +92 11% |
| walker, stand | 484 ±185 | 908 ±38 | 924 ±28 | 928 ±50 | 947 ±14 | **949** ±10 | **961** ±19 | +12 1% |
| walker, walk | 325 ±26 | 720 ±69 | 518 ±92 | 691 ±120 | 662 ±75 | **830** ±58 | **904** ±34 | +74 9% |
| cheetah, run | 179 ±65 | 241 ±25 | 215 ±15 | 278 ±51 | 253 ±27 | **308** ±34 | **348** ±28 | +40 13% |

| DMControl (video-hard) | SAC | DrQ | SODA | SVEA (overlay) | SRM | SGQN | SMG (ours) | Δ |
|---|---|---|---|---|---|---|---|---|
| cartpole, swingup | 156 ±16 | 168 ±35 | 346 ±59 | 510 ±177 | 254 ±69 | **599** ±112 | **764** ±32 | +165 28% |
| finger, spin | 22 ±10 | 54 ±44 | 310 ±72 | 353 ±71 | 131 ±89 | **710** ±159 | **910** ±61 | +200 28% |
| walker, stand | 212 ±41 | 278 ±79 | 406 ±68 | 814 ±57 | 558 ±139 | **870** ±78 | **955** ±9 | +85 10% |
| walker, walk | 132 ±26 | 110 ±33 | 175 ±31 | 348 ±80 | 165 ±99 | **634** ±136 | **814** ±51 | +180 28% |
| cheetah, run | 56 ±30 | 38 ±26 | 118 ±40 | 105 ±13 | 87 ±24 | **135** ±44 | **303** ±46 | +168 124% |

Table 1 reports the generalization performance of SMG and all baseline methods with the video-background modification, which is the most challenging evaluation setting. The table shows that SMG outperforms all baselines in all ten tasks. Particularly impressive is SMG's superiority in *video-hard*; when removing the floor and the walker's shadow, the performance of all baseline methods drops significantly. However, SMG is less affected by this substantial distribution shift and maintains a stable performance across all tasks, with episode returns boosted more than 160 over the second-best in four out of five tasks (as *walker-stand* is a much easier task to train), showcasing its exceptional generalization capability.

### 4.3 Robotic Manipulation Results

To further validate SMG's applicability to more realistic tasks, we conduct experiments on two goal-reaching robotic manipulation tasks [17], including *peg-in-box* and *reach*, and following similar generalization settings used in [3]. As illustrated in Figure 6, there are five different testing settings with different colors and textures for the background and the table. We train all methods for 250k steps and use random convolutions [22] as the data augmentation for baseline methods, as it aligns better with the testing scenarios. SMG continued to use hybrid augmentation as previously mentioned.

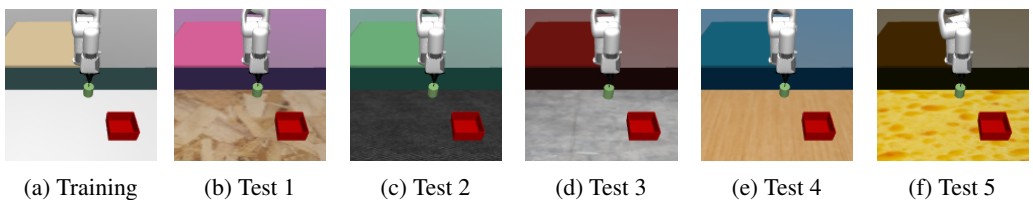

(a) Training     (b) Test 1     (c) Test 2     (d) Test 3     (e) Test 4     (f) Test 5

Figure 6: Examples of training and testing observation for the robotic environment (*Peg-in-box*). (b-f) indicates five different evaluation settings varying in background colors and table textures.

Table 2 presents the evaluation results for *peg-in-box*, a task where a robot must insert a peg tied to its arm into a box. SMG achieves dominant performance across all evaluation settings, boosting an average improvement of $102\%$ over the second-best method. Impressively, SMG exhibits remarkable stability across the six evaluation settings, with a standard deviation of only 7, while baseline methods all fail in some evaluation settings. This underscores SMG's generalization capability. These results also highlight SMG's superiority in realistic tasks, as its reconstruction-based auxiliary loss can capture more detailed features in the image, which is hard for methods that mainly rely on data augmentation techniques.

Table 2: Robotic manipulation results in *peg-in-box*. Red indicates the best and blue indicates the second-best. $\Delta$ = improvement of SMG over the second best. The last row reports the average performance over all six evaluation settings.

| Robtic-Manipulation (*peg-in-box*) | SAC | DrQ | SODA | SVEA (overlay) | SRM | SGQN | SMG (ours) | $\Delta$ |
|---|---|---|---|---|---|---|---|---|
| train | 31 ±73 | **233** ±14 | 232 ±20 | 212 ±39 | 227 ±15 | 232 ±19 | **237** ±16 | +4 2% |
| test1 | −33 ±25 | **63** ±99 | 34 ±143 | −18 ±59 | 55 ±98 | −67 ±28 | **237** ±18 | +174 276% |
| test2 | −42 ±31 | −40 ±77 | 76 ±119 | 85 ±68 | 11 ±54 | **194** ±51 | **219** ±37 | +25 13% |
| test3 | −8 ±46 | 15 ±107 | 66 ±147 | 67 ±73 | 147 ±114 | **198** ±34 | **237** ±15 | +39 20% |
| test4 | −42 ±51 | 72 ±28 | 80 ±122 | 109 ±98 | **112** ±123 | −51 ±46 | **237** ±17 | +125 112% |
| test5 | −52 ±31 | −54 ±30 | −104 ±51 | −26 ±102 | **143** ±122 | −108 ±24 | **237** ±15 | +94 66% |
| **Average** | −24 ±28 | 48 ±95 | 64 ±98 | 72 ±80 | **116** ±69 | 66 ±143 | **234** ±7 | +118 102% |

## 4.4 Ablation Study

In order to explore the role played by different loss terms in SMG, we conduct an ablation study in DMControl tasks. Table 3 presents the performance drop without each loss term compared to the full model in the *video-hard* setting. The results indicate that every loss term contributes significantly to the final performance. Notably, $L_{q\_consist}$ exhibits the most substantial impact on performance, highlighting the importance of maintaining stable Q-value estimation in generalization tasks. Moreover, the performance drop without $L_{back}$ or $L_{mask}$ is around $20\%$ to $30\%$, underlining the importance of attribution augmentation in enhancing SMG's generalization in video-background settings, as the two loss terms directly affect the quality of the attribution augmentation. Additionally, $L_{action}$ aids in learning a better task-relevant representation. As for $L_{fore\_consist}$, it also contributes to improving generalization ability, particularly in relatively challenging tasks where the performance improvement ranges from $15\%$ to $25\%$.

Table 3: Ablation study in DMControl (*video-hard*). Red indicates the performance drop of the ablated model compared to the full model.

| DMControl (video hard) | SMG (full) | w/o $L_{fore\_consis}$ | w/o $L_{action}$ | w/o $L_{back}$ | w/o $L_{mask}$ | w/o $L_{q\_consis}$ |
|---|---|---|---|---|---|---|
| cartpole, swingup | 764 ± 32 | 720 ± 100 −44 (6%) | 631 ± 92 −133 (17%) | 763 ± 44 −1 (0%) | 590 ± 84 −174 (23%) | 302 ± 30 −462 (60%) |
| finger, spin | 910 ± 61 | 695 ± 103 −215 (24%) | 609 ± 352 −301 (33%) | 412 ± 170 −498 (55%) | 731 ± 130 −179 (20%) | 509 ± 83 −401 (44%) |
| walker, stand | 955 ± 9 | 885 ± 45 −70 (7%) | 855 ± 96 −100 (10%) | 775 ± 144 −180 (19%) | 836 ± 127 −119 (12%) | 432 ± 210 −523 (55%) |
| walker, walk | 814 ± 51 | 642 ± 63 −172 (21%) | 670 ± 22 −144 (18%) | 657 ± 103 −157 (19%) | 416 ± 98 −398 (49%) | 282 ± 34 −532 (65%) |
| cheetah, run | 303 ± 46 | 247 ± 40 −56 (18%) | 212 ± 52 −91 (30%) | 233 ± 110 −70 (23%) | 162 ± 100 −141 (47%) | 130 ± 37 −173 (57%) |
| **Average** | | −15% | −22% | −23% | −30% | −56% |

To better grasp the significance of $L_{\text{mask}}$ and $L_{\text{back}}$ in SMG, we showcase the predicted masks and their corresponding attribution augmentations in Figure 7. When $L_{\text{mask}}$ is removed, the model generates an almost white mask, indicating that the foreground model overly captures irrelevant features without the constraint of mask ratio loss. Consequently, only a few parts are replaced by a random image in the attribution augmentation. In contrast, removing $L_{\text{back}}$ causes the background model to learn all features excessively, resulting in attribution augmentation images devoid of task-relevant information. The ablation results underscore that both $L_{\text{mask}}$ and $L_{\text{back}}$ are vital in crafting meaningful attribution augmentations, which in turn are utilized by the two consistency losses and impact the representation learning process. We conduct more experiments in Appendix E to reveal that $L_{\text{mask}}$ serves as a guiding factor in mask learning and SMG is not significantly influenced by variations in the hyperparameter mask ratio $\rho$.

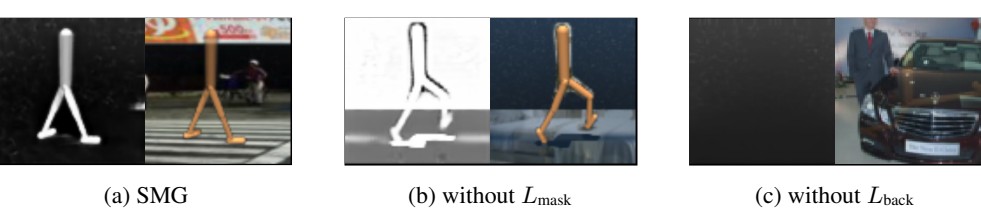

(a) SMG          (b) without $L_{\text{mask}}$          (c) without $L_{\text{back}}$

Figure 7: Predicted masks and corresponding attribution augmentations. (a) is the full model, (b) and (c) are the models without $L_{\text{mask}}$ and $L_{\text{mask}}$ respectively.

## 5 Related Work

**Improving generalization ability of RL agents** has drawn increasing attention in recent years. Researchers primarily explore two aspects: using data augmentation techniques to inject useful priors when training [20, 15, 16, 22, 14, 32, 38] and employing various auxiliary tasks to guide the learning process [13, 3, 1, 42, 40, 12]. For example, Hansen and Wang [12] regularize the representations between observations with its augmented view through an auxiliary prediction task; Hansen et al. [14] stabilize Q-values via delicately design the data augmentation process; Bertoin et al. [3] introduce saliency maps to visualize the focus of Q-functions; Wang et al. [40] extract the foreground objects by employing a segment anything model. Orthogonal to existing works, we argue that focusing the RL agent on task-relevant features across diverse deployment scenarios can substantially boost the generalization capability. We propose a novel reconstruction-based auxiliary task to achieve this goal.

**Decision-making based on task-relevant features** can substantially enhance the performance and robustness of RL agents [4, 45, 43, 29]. Bharadhwaj et al. [4] use an empowerment term to distill control-relevant features from the task; Zhu et al. [45] bolster the resilience of RL agents by regularizing the posterior predictability; Zhang et al. [43] learns compact representations by bisimulation metrics. Additionally, methods utilizing separated model architectures to extract different types of features simultaneously have been proposed [6, 39, 30, 25, 37]. For instance, Wang et al. [39] decompose the latent state into four parts based on their interaction with actions and rewards; Pan et al. [30] leverage both controllable and non-controllable states in policy learning; Wan et al. [37] apply task-relevant features to imitation learning. Our work also employs separated models. However, we prudently design this architecture in a model-free setting and propose novel loss terms to enhance the accuracy of image predictions.

A detailed comparison between SMG and other methods is provided in Appendix F.2.

## 6 Conclusion and Future Work

In this paper, we propose SMG for visual-based RL generalization and show its superiority in sample efficiency, stability, and generalization through extensive experiments. The success of SMG can be attributed to two key factors: (i) a delicately designed reconstruction-based auxiliary task with separated models architecture, which enables the RL agent to extract task-relevant and task-irrelevant representations from visual observations simultaneously; (ii) two consistency losses to further guide the RL agent's focus under deployment scenarios. We believe that the proposed method can be applied to a wide range of tasks.

SMG is particularly well-suited for robotic manipulation tasks in realistic scenarios. However, when the observation contains too many task-relevant objects, the complexity of accurately learning a mask increases. This can lead to a decline in SMG's performance. For instance, in an autonomous navigation task, the presence of numerous pedestrians in the view makes it challenging to accurately mask all of them.

The future work includes exploring more advanced backbones for task-relevant feature extraction, taking into account the generalization on non-static camera viewpoints and the test of SMG on realistic tasks to verify its generalization ability in real applications.

## Acknowledgments and Disclosure of Funding

This work is supported by the National Key Research and Development Program of China (No. 2020YFA0711402).

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

## A  Derivations

We formulate the representation learning objective as a variational lower bound of the mutual information [31, 24] between the observation $o_t$ and the representation $z_t$. By considering the independence between the task-relevant and task-irrelevant representations, we can decompose the mutual information as:

$$
\begin{aligned}
I(o_t; z_t) &= \mathbb{E}_{p(o_t, z_t)}[\log p(o_t|z_t) - \log p(o_t)] \\
&\geq \mathbb{E}_{p(o_t, z_t)}[\log p(o_t|z_t)] \\
&\geq \mathbb{E}_{p(o_t, z_t)}[\log p(o_t|z_t)] - \mathbb{E}_{p(z_t)}[\mathbb{D}_{KL}(p(o_t|z_t)||q(o_t|z_t))] \\
&= \mathbb{E}_{p(z_t, o_t)}[\log q(o_t|z_t)] \\
&= \mathbb{E}_{q(z_t|o_t)p(o_t)}[\log q(o_t|z_t)] \\
&= \mathbb{E}_{q(z_t^+|o_t)q(z_t^-|o_t)p(o_t)}[\log q(o_t|z_t^+, z_t^-)] \\
&= \mathbb{E}_{o_t \sim \mathcal{D}}[\mathbb{E}_{z_t^+ \sim f^+(o_t), z_t^- \sim f^-(o_t)}[\log q(o_t|z_t^+, z_t^-)]]
\end{aligned}
\tag{11}
$$

We use the empowerment term $I(a_t, z_{t+1}^+|z_t^+)$ introduced in [28] to quantify the information contained in the representation $z_{t+1}^+$ about the selected action $a_t$, in goal of enhance the control-relevant property of the task-relevant representation $z_t^+$. We derive the variational lower bound of the empowerment term as:

$$
\begin{aligned}
I(a_t, z_{t+1}^+|z_t^+) &= \mathbb{E}_{p(a_t, z_{t+1}^+, z_t^+)}[\log \frac{p(a_t|z_{t+1}^+, z_t^+)}{p(a_t|z_t^+)}] \\
&= \mathbb{E}_{p(a_t, z_{t+1}^+, z_t^+)}[\log \frac{q(a_t|z_{t+1}^+, z_t^+)}{p(a_t|z_t^+)} + \log \frac{p(a_t|z_{t+1}^+, z_t^+)}{q(a_t|z_{t+1}^+, z_t^+)}] \\
&\geq \mathbb{E}_{p(a_t, z_{t+1}^+, z_t^+)}[\log \frac{q(a_t|z_{t+1}^+, z_t^+)}{p(a_t|z_t^+)}] \\
&= \mathbb{E}_{p(a_t, z_{t+1}^+, z_t^+)}[\log q(a_t|z_{t+1}^+, z_t^+)] - \int p(z_t^+)p(a_t|z_t^+)p(z_{t+1}^+|z_t^+, a_t) \log p(a_t|z_t^+) \\
&= \mathbb{E}_{p(a_t, z_{t+1}^+, z_t^+)}[\log q(a_t|z_{t+1}^+, z_t^+)] + \mathbb{E}_{p(z_t^+)p(z_{t+1}^+|z_t^+, a_t)}[H(p(a_t|z_t^+))] \\
&\geq \mathbb{E}_{p(a_t, z_{t+1}^+, z_t^+)}[\log q(a_t|z_{t+1}^+, z_t^+)]
\end{aligned}
\tag{12}
$$

In practice, we integrate a parameterized inverse dynamic model to predict the action $a_t$ based on the two continuous representations $z_t^+$ and $z_{t+1}^+$. We employ the Mean Squared Error (MSE) loss to guide the training of the inverse dynamic model.

# B   Pseudocode

---

**Algorithm 1** SAC with Separated Models

---

**Denote** network parameters $\theta$, mask ratio $\rho$, batch size $N$, replay buffer $\mathcal{B}$
**Denote** policy network $\pi_\theta$, foreground encoder $f_\theta^+$, background encoder $f_\theta^-$
**foreach** *iteration time step* **do**

$\quad$ $a, o', r \sim \pi_\theta(f_\theta^+(o)), \mathcal{P}(o, a), \mathcal{R}(o, a)$
$\quad$ $\mathcal{B} \leftarrow \mathcal{B} \cup (o, a, r, o')$
$\quad$ **foreach** *update time step* **do**

$\quad\quad$ $\{o_i, a_i, r_i, o'_i\}_{i \in [1, N]} \sim \mathcal{B}$
$\quad\quad$ $o_i^+, mask_i \sim f_\theta^+(o_i)$
$\quad\quad$ $o_i^- \sim f_\theta^-(o_i)$
$\quad\quad$ $o_i^{aug} \leftarrow o_i^+ * mask_i + \epsilon * (1 - mask_i)$  // $\epsilon$ is sampled from image dataset
$\quad\quad$ $L_{recon} \leftarrow L(o_i, o_i^+ * mask_i + o_i^- * (1 - mask_i))$  // Equation 2
$\quad\quad$ $L_{fore\_consist} \leftarrow L(o_i^+, f_\theta^+(o_i^{aug}))$  // Equation 7
$\quad\quad$ $L_{back} \leftarrow L(\epsilon, f_\theta^-(o_i^{aug}))$  // Equation 4
$\quad\quad$ $L_{action} \leftarrow L(o_i, o'_i, a)$  // Equation 6
$\quad\quad$ $L_{mask} \leftarrow L(mask_i, \rho)$  // Equation 3
$\quad\quad$ $L_{q\_consist} \leftarrow L(Q_\theta(f_\theta^+(o_i), a), Q_\theta(f_\theta^+(o_i^{aug}), a))$  // Equation 8
$\quad\quad$ $L_{aux} \leftarrow L_{recon} + L_{fore\_consist} + L_{back} + L_{action} + L_{mask}$  // auxiliary loss
$\quad\quad$ $L_{critic} \leftarrow L_q + L_{q\_consist}$  // critic loss
$\quad\quad$ **update** $\theta$ with $L_{actor}, L_{critic}, L_{aux}$

$\quad$ **end for**

**end for**
$L_q, L_{actor}$ are defined by SAC

---

# C   More Experiment Details

## C.1   Computing Hardware

We conduct all experiments on a single machine equipped with an AMD EPYC 7B12 CPU (64 cores), 512GB RAM, and eight NVIDIA GeForce RTX 3090 GPUs (24 GB memory). We report the training wall time of different methods on DMControl tasks in Table 4.

Table 4: Wall time comparison of different methods on DMControl tasks.

| Algorithm | Wall Time (DMControl, 500k) |
|-----------|------------------------------|
| SAC | $\sim 10$ hours |
| DrQ | $\sim 13$ hours |
| SODA | $\sim 12$ hours |
| SVEA | $\sim 12$ hours |
| SRM | $\sim 8$ hours |
| SGQN | $\sim 12$ hours |
| SMG (ours) | $\sim 22$ hours |

## C.2   Network Architecture

We reproduce all baseline methods with the official code of DMC-GB (`https://github.com/nicklashansen/dmcontrol-generalization-benchmark`) published by Nicklas Hansen, and we build our model on top of the SAC implementation. We use the same encoder and decoder architecture as the baseline methods to ensure a fair comparison.

Figure 8 provides a detailed view of the encoder and decoder architecture. The input observation shape is $9 \times 84 \times 84$, achieved by stacking three continuous frames. The encoder network contains 12 stacked convolutional layers, each with 32 filters of size $3 \times 3$. The stride is set to 1 for the first layer and 2 for the subsequent ones, facilitating down-sampling of the visual input. Then, after a flatten operation and a fully connected layer, an embedding of size *embedding_size* $\times 1$ is obtained. Before decoding, SMG first expands the embedding into triples of the same size, aiming to decode three stacked input images separately. These three embeddings are then individually fed into the same decoder network, which consists of two groups of convolutional and upsampling layers to reconstruct the observation. The foreground decoder outputs the reconstructed foreground and a mask, while the background decoder outputs only the reconstructed background. For the inverse dynamic model, we adopt the architecture from [13], which utilizes multi-layer perceptions to project the concatenation of two embeddings into the action space.

The number of parameters in SMG is approximately double that of the baseline methods due to the use of two model branches. However, the performance improvement is primarily due to the novel model architecture rather than the increase in the number of parameters, as we use encoder and decoder networks similar to those in the baseline methods.

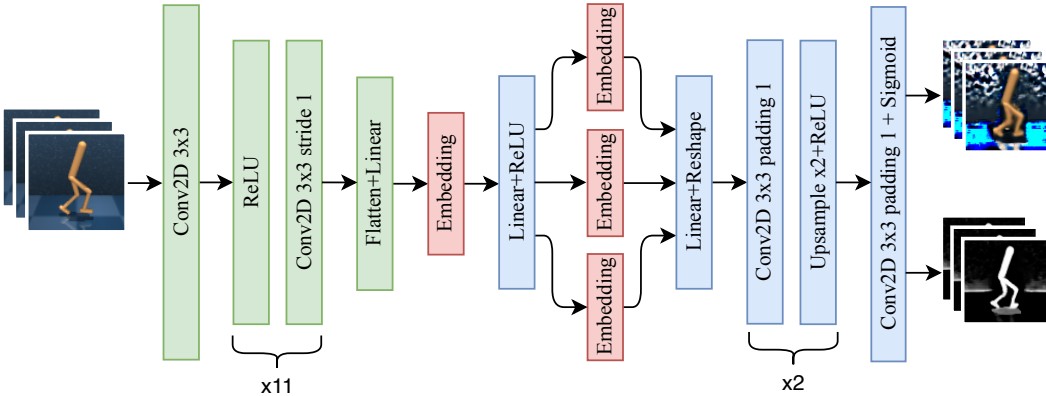

Figure 8: SMG network architecture (foreground encoder + foreground decoder).

## C.3 Hyperparameters

We report the hyperparameters used in our experiments in Table 5. We use the same hyperparameters for all seven tasks, except the action repeat and the mask ratio $\rho$. The $L_{\text{aux}}$ in SMG comprises five loss terms, which seems challenging to balance the weights. However, through experiments, we found that setting average weights for $L_{recon}, L_{mask}, L_{action}, L_{back}$ is sufficient to achieve good performance (except the $\lambda_{back}$ is set to 2 since the background model should train to fit more complex images). Regarding the $L_{fore}$, a too-large weight would lead to the model overfitting the inaccurate attribution predictions in the early stage (as we use the model output under raw observation as ground truth), so we set it to 0.1.

## D  More Experiment Results

### D.1  Training Curves

We present the training curves for all seven tasks in Figure 11, including four evaluation settings of DMControl and Robotic Manipulation tasks. As depicted in the figure, SMG demonstrates notably faster convergence and higher asymptotic performance across nearly all training and evaluation settings, showcasing the effectiveness of the reconstruction-based auxiliary task in enhancing sample efficiency. SMG exhibits superiority, particularly in the *video-hard* setting of DMControl tasks, where the performance of other methods drops evidently when random videos replace the background. Additionally, the figure underscores the considerable challenge posed by Robotic Manipulation tasks, with only SMG and SGQN successfully achieving zero-shot generalization in evaluation settings.

Table 5: Hyperparameters.

| Hyperparameter | Value |
|---|---|
| Observation size | $84 \times 84$ |
| Frame stack | 3 |
| Discount factor $\gamma$ | 0.99 |
| Batch size | 64 |
| Embedding size | 256 |
| Action repeat | 8 (*cartpole-swingup*), 4 (*walker-walk*, *walker-stand*, *cheetah-run*) |
| | 2 (*finger-spin*), 1 (*reach*, *peg-in-box*) |
| Train steps | 250k (*walker-stand*, *reach*, *peg-in-box*), 500k (others) |
| Replay buffer size | 500k |
| Actor optimizer | Adam ($lr = 1e-3, \beta_1 = 0.9, \beta_2 = 0.999$) |
| Critic optimizer | Adam ($lr = 1e-3, \beta_1 = 0.9, \beta_2 = 0.999$) |
| Auxiliary task optimizer | Adam ($lr = 1e-3, \beta_1 = 0.9, \beta_2 = 0.999$) |
| Auxiliary task update frequency | 2 |
| Reconstruction loss weight $\lambda_{recon}$ | 1 |
| Background reconstruction loss weight $\lambda_{back}$ | 2 |
| Mask ratio loss weight $\lambda_{mask}$ | 1 |
| Empowerment loss weight $\lambda_{action}$ | 1 |
| Q-value consistency loss weight $\lambda_{q\_consist}$ | 0.5 |
| Foreground consistency loss weight $\lambda_{fore\_consist}$ | 0.1 |
| Mask ratio $\rho$ | 0.12 (*reach*, *peg-in-box*), 0.06 (*walker-walk*, *walker-stand*, *cheetah-run*) |
| | 0.04 (*cartpole-swingup*, *finger-spin*) |

Moreover, SMG shows more stable performance across different evaluation settings, which is crucial for real-world applications.

## D.2    More Table Results

Table 6: DMControl results in random-color settings.

| DMControl-GB (*color-easy*) | SAC | DrQ | SODA | SVEA (overlay) | SRM (SAC) | SGQN | SMG (ours) | Δ |
|---|---|---|---|---|---|---|---|---|
| cartpole, swingup | 178 ±24 | 845 ±29 | 720 ±109 | 809 ±40 | **856** ±14 | 764 ±84 | **854** ±13 | −2 0% |
| finger, spin | 296 ±22 | 827 ±174 | 761 ±87 | **919** ±43 | 916 ±34 | 852 ±126 | **957** ±52 | +38 4% |
| walker, stand | 592 ±274 | 827 ±97 | 929 ±23 | **957** ±4 | 953 ±5 | 906 ±50 | **965** ±13 | +8 1% |
| walker, walk | 430 ±33 | 669 ±68 | 539 ±51 | 705 ±124 | 632 ±93 | **805** ±47 | **915** ±36 | +110 14% |
| cheetah, run | 253 ±27 | 237 ±74 | 219 ±46 | 289 ±43 | 272 ±24 | **312** ±34 | **346** ±27 | +34 11% |

| DMControl-GB (*color-hard*) | SAC | DrQ | SODA | SVEA (overlay) | SRM (SAC) | SGQN | SMG (ours) | Δ |
|---|---|---|---|---|---|---|---|---|
| cartpole, swingup | 184 ±26 | 717 ±133 | 585 ±66 | **752** ±86 | **752** ±103 | 636 ±110 | 726 ±62 | −26 3% |
| finger, spin | 271 ±23 | 655 ±214 | 663 ±106 | **868** ±74 | 834 ±90 | 700 ±219 | **841** ±113 | −27 3% |
| walker, stand | 526 ±259 | 769 ±182 | 719 ±138 | 799 ±118 | **807** ±128 | 788 ±114 | **878** ±70 | +71 9% |
| walker, walk | 379 ±37 | 456 ±192 | 396 ±78 | 571 ±134 | 483 ±123 | **632** ±176 | **739** ±31 | +107 17% |
| cheetah, run | 208 ±54 | 147 ±80 | 199 ±38 | **238** ±69 | 203 ±30 | 210 ±18 | **299** ±22 | +61 26% |

Table 6 shows the generalization performance of SMG and all baseline methods with the random-color modification in DMControl tasks. SMG outperforms all baselines in 7 out of 10 tasks, with the performance gap within 5% in the other three tasks. The results indicate that SMG not only performs well in video-background settings but also exhibits superior generalization capability in random-color settings. This is achieved because overlaying the observation with random images can also introduce color shift.

Table 7: Training and average performance in DMControl.

| DMControl-GB (training) | SAC | DrQ | SODA | SVEA (overlay) | SRM (SAC) | SGQN | SMG (ours) | Δ |
|---|---|---|---|---|---|---|---|---|
| cartpole, swingup | 186 ±6 | **872** ±10 | 687 ±175 | 809 ±42 | **871** ±10 | 805 ±58 | 858 ±9 | −14 2% |
| finger, spin | 306 ±12 | 884 ±115 | 801 ±65 | 923 ±36 | **925** ±35 | 922 ±61 | **961** ±44 | +36 4% |
| walker, stand | 630 ±224 | 955 ±18 | 881 ±51 | **959** ±5 | 959 ±6 | 952 ±17 | **964** ±18 | +5 1% |
| walker, walk | 422 ±42 | 827 ±61 | 581 ±129 | 753 ±143 | 715 ±74 | **876** ±45 | **924** ±31 | +48 5% |
| cheetah, run | 311 ±36 | 333 ±43 | 225 ±39 | 300 ±37 | 298 ±30 | **343** ±37 | **357** ±25 | +14 4% |

| DMControl-GB (average) | SAC | DrQ | SODA | SVEA (overlay) | SRM (SAC) | SGQN | SMG (ours) | Δ |
|---|---|---|---|---|---|---|---|---|
| cartpole, swingup | 176 ±11 | 642 ±255 | 591 ±132 | **720** ±110 | 676 ±226 | 704 ±77 | **808** ±53 | +88 12% |
| finger, spin | 213 ±107 | 586 ±297 | 630 ±173 | 776 ±215 | 690 ±297 | **809** ±88 | **924** ±45 | +115 14% |
| walker, stand | 489 ±147 | 747 ±243 | 772 ±198 | 891 ±70 | 845 ±154 | **893** ±61 | **945** ±33 | +52 6% |
| walker, walk | 338 ±109 | 556 ±254 | 442 ±147 | 614 ±146 | 531 ±199 | **755** ±103 | **859** ±72 | +104 14% |
| cheetah, run | 201 ±85 | 199 ±100 | 195 ±40 | 242 ±72 | 223 ±75 | **262** ±77 | **331** ±24 | +69 26% |

For a more direct measurement of the generalization ability in DMControl, we further calculate the average performance across five evaluation settings (including performance under training observation) and report the results in Table 7. As shown in the table, SMG achieves state-of-the-art zero-shot generalization capability in all five DMControl tasks, surpassing all baseline methods by a margin of up to 26%. The results also demonstrate SMG's stability across different evaluation settings, with standard deviations less than 80 in all tasks. In contrast, the standard deviations of other methods range from 100 to 250.

Table 8: Robotic manipulation results in *reach*.

| Robtic-Manipulation (*Reach*) | SAC | DrQ | SODA | SVEA (overlay) | SRM (SAC) | SGQN | SMG (ours) | Δ |
|---|---|---|---|---|---|---|---|---|
| train | 4 ±18 | 32 ±3 | 11 ±14 | **33** ±2 | 30 ±2 | **33** ±2 | 30 ±2 | −3 9% |
| test1 | −16 ±33 | −1 ±23 | −26 ±9 | −22 ±16 | −3 ±25 | **19** ±13 | **30** ±1 | +11 58% |
| test2 | −10 ±22 | −9 ±11 | −17 ±16 | −21 ±22 | −8 ±22 | **33** ±2 | **24** ±6 | −9 27% |
| test3 | −32 ±14 | −38 ±29 | −20 ±34 | −13 ±10 | 24 ±9 | **33** ±2 | **30** ±2 | −3 9% |
| test4 | −19 ±50 | 10 ±26 | −21 ±16 | 0 ±21 | −1 ±30 | **24** ±6 | **29** ±1 | +5 21% |
| test5 | −54 ±11 | −33 ±19 | −50 ±7 | −37 ±27 | **-8** ±29 | −16 ±22 | **29** ±2 | +37 462% |
| **Average** | −21 ±18 | −6 ±24 | −20 ±18 | −10 ±22 | 6 ±15 | **21** ±17 | **29** ±2 | +8 38% |

The experiment results of robotic manipulation *reach* are reported in Table 8. SMG also shows a stable and superior performance in this task, with an average improvement of 38% over the second-best method.

# E    More Ablation Study

We report the effect of removing each loss term to the average performance across five evaluation settings in DMControl tasks in Table 9. Compared with Table 3, $L_{q\_consist}$ still exhibits the most substantial impact on performance, though the performance drop is slightly smaller. This may be because the random-color settings do not shift the observations heavily compared to the video-background settings, so the Q-value estimation is less affected. A similar phenomenon is observed in $L_{back}$ and $L_{mask}$, indicating that attribution augmentation is more crucial in video-background settings.

The mask ratio $\rho$ is a hyperparameter that controls the expected proportion of the foreground area. However, this parameter is an empirical choice and may not precisely match the actual proportion of a given task. To investigate the sensitivity of SMG to the mask ratio, we conduct experiments with different $\rho$ values in the *walker-walk* task of the *video-hard* setting. We select $\rho$ values ranging from 0.02 to 0.1 with an interval of 0.02 and report the average performance across five evaluation settings in Figure 9. The results indicate that variations do not significantly influence SMG in the mask ratio, as $\rho$ values between 0.04 and 0.08 achieve similar performance. Moreover, when $\rho$ is too small (0.02) or too large (0.1), the performance drops around 6% compared to the optimal $\rho$ value (0.06). We also report the predicted masks of different $\rho$ values in the figure. As $\rho$ increases, the predicted masks start to include background areas, so a too high value leads to decreased performance. Conversely, when $\rho$ is too small, the mask depicts an inaccurate foreground area (e.g. the legs of the walker with $\rho = 0.02$), resulting in a performance drop as well.

Table 9: Ablation study in DMControl (average performance).

| DMControl (average) | SMG (full) | w/o $L_{\text{fore\_consis}}$ | w/o $L_{\text{action}}$ | w/o $L_{\text{back}}$ | w/o $L_{\text{mask}}$ | w/o $L_{\text{q\_consis}}$ |
|---|---|---|---|---|---|---|
| cartpole, swingup | $808 \pm 53$ | $763 \pm 28$ $-45\ (6\%)$ | $762 \pm 71$ $-46\ (6\%)$ | $795 \pm 32$ $-13\ (2\%)$ | $758 \pm 97$ $-50\ (6\%)$ | $646 \pm 191$ $-162\ (20\%)$ |
| finger, spin | $924 \pm 45$ | $815 \pm 66$ $-109\ (12\%)$ | $791 \pm 112$ $-133\ (14\%)$ | $640 \pm 115$ $-284\ (31\%)$ | $866 \pm 73$ $-58\ (6\%)$ | $773 \pm 151$ $-151\ (16\%)$ |
| walker, stand | $945 \pm 33$ | $918 \pm 26$ $-27\ (3\%)$ | $874 \pm 17$ $-71\ (8\%)$ | $915 \pm 70$ $-30\ (3\%)$ | $930 \pm 47$ $-15\ (2\%)$ | $598 \pm 114$ $-347\ (37\%)$ |
| walker, walk | $859 \pm 72$ | $727 \pm 54$ $-132\ (15\%)$ | $757 \pm 61$ $-102\ (12\%)$ | $756 \pm 103$ $-103\ (12\%)$ | $693 \pm 145$ $-166\ (19\%)$ | $613 \pm 227$ $-246\ (29\%)$ |
| cheetah, run | $331 \pm 24$ | $304 \pm 29$ $-27\ (8\%)$ | $325 \pm 58$ $-6\ (2\%)$ | $270 \pm 25$ $-61\ (18\%)$ | $319 \pm 82$ $-12\ (4\%)$ | $269 \pm 110$ $-62\ (19\%)$ |
| **Average** | | $-9\%$ | $-8\%$ | $-13\%$ | $-7\%$ | $-24\%$ |

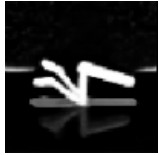 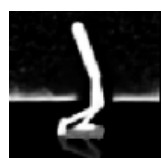 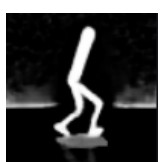 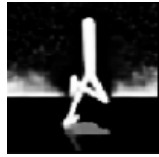 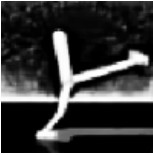

(a) $\rho = 0.02$ (800)    (b) $\rho = 0.04$ (857)    (c) $\rho = 0.06$ (859)    (d) $\rho = 0.08$ (823)    (e) $\rho = 0.1$ (793)

Figure 9: Ablation study of mask ratio $\rho$ in *walker-walk* of average performance across five evaluation settings. The images and numbers in parentheses indicate the predicted masks and the corresponding performance, respectively.

# F    More Discussion

## F.1    Bootstrapping Process in SMG

The attribution augmentation utilized in SMG requires the model to predict an accurate mask, and the foreground consistency loss also requires a precise attribution prediction of the model. This might seem contradictory, as the model struggles to make meaningful predictions in the early stages, which means it cannot satisfy the two requirements immediately. We dig into the training process of SMG

by experiments and provide the model outputs in different training stages in Figure 10. In the very early stage ($\leq 1000$ steps), the model has difficulty predicting accurate masks, leading the attribution augmentation more likes an overlay augmentation. However, the model rapidly learns to predict relatively accurate masks and generate meaningful attribution augmentation images that can help optimize $L_{\text{back}}$ and $L_{\text{fore\_consist}}$ (after 2000 steps), aided by the constraint of $L_q$. Subsequently, with the inclusion of $L_{\text{back}}$ and $L_{\text{fore\_consist}}$, the network begins to focus more on task-relevant areas in the observation, thereby in turn comes back to enhance the accuracy of Q-values and foreground predictions. Consequently, we view the training of SMG as a bootstrapping process.

## F.2  Comparison with Related Work

TIA [6] also designs two model branches to capture task and distractor features, similar to our separated models architecture. However, SMG differs from TIA in several essential aspects: (i) TIA is a model-based method focusing on eliminating task-irrelevant distractors in training observations, while SMG aims to utilize task-relevant features across diverse deployment scenarios to enhance the generalization capability of RL agents; (ii) SMG operates in a model-free setting, which can be more efficient to train and more flexible for applying data augmentation techniques; (iii) TIA uses a background-only reconstruction loss and requires the background model to reconstruct the full observation, which may cause the background branch to overly fit task-relevant features. In contrast, SMG addresses this issue by introducing attribution augmentation images to supervise the background model; (iv) SMG utilizes mask ratio loss to learn a more precise mask, while the masks in TIA are prone to containing distractors, as reported in its original paper.

SODA [12] also improves the generalization ability of RL agents by regularizing the representations between observations and their augmented views, similar to the consistency losses in SMG. However, SODA implements this by simply minimizing the L2 distance between the two representations, which imposes a too rigid constraint and lacks interpretability. We achieve this by introducing Q-value consistency loss and foreground consistency loss, which provide more explainable supervision and additionally improve the stability of Q-values and predicted attributions.

Note that the core idea underlying the Q-value loss in Equation 8 differs significantly from the consistency regulation objective proposed by SGQN [3]. SGQN focuses on prioritizing pixels that belong to the saliency map during encoding, primarily to enhance the accuracy of Q-value estimation under raw observations. In contrast, SMG treats the Q-values under raw observations as the ground truth and aims to achieve consistency between these Q-values and those obtained under augmented observations. Thus, we additionally use a stop-gradient operation.

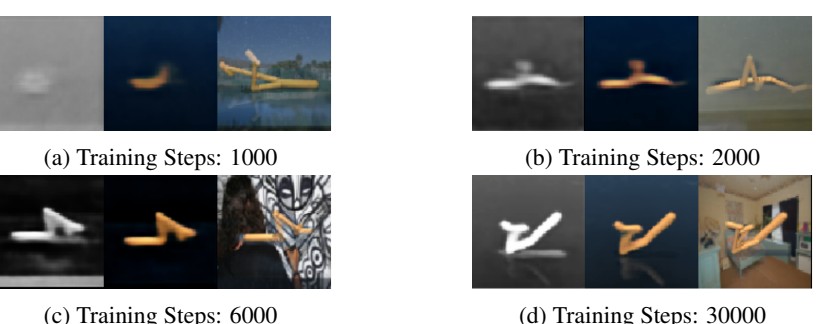

(a) Training Steps: 1000  (b) Training Steps: 2000

(c) Training Steps: 6000  (d) Training Steps: 30000

Figure 10: Masks, attributions, and corresponding attribution augmentation images in different training stages.

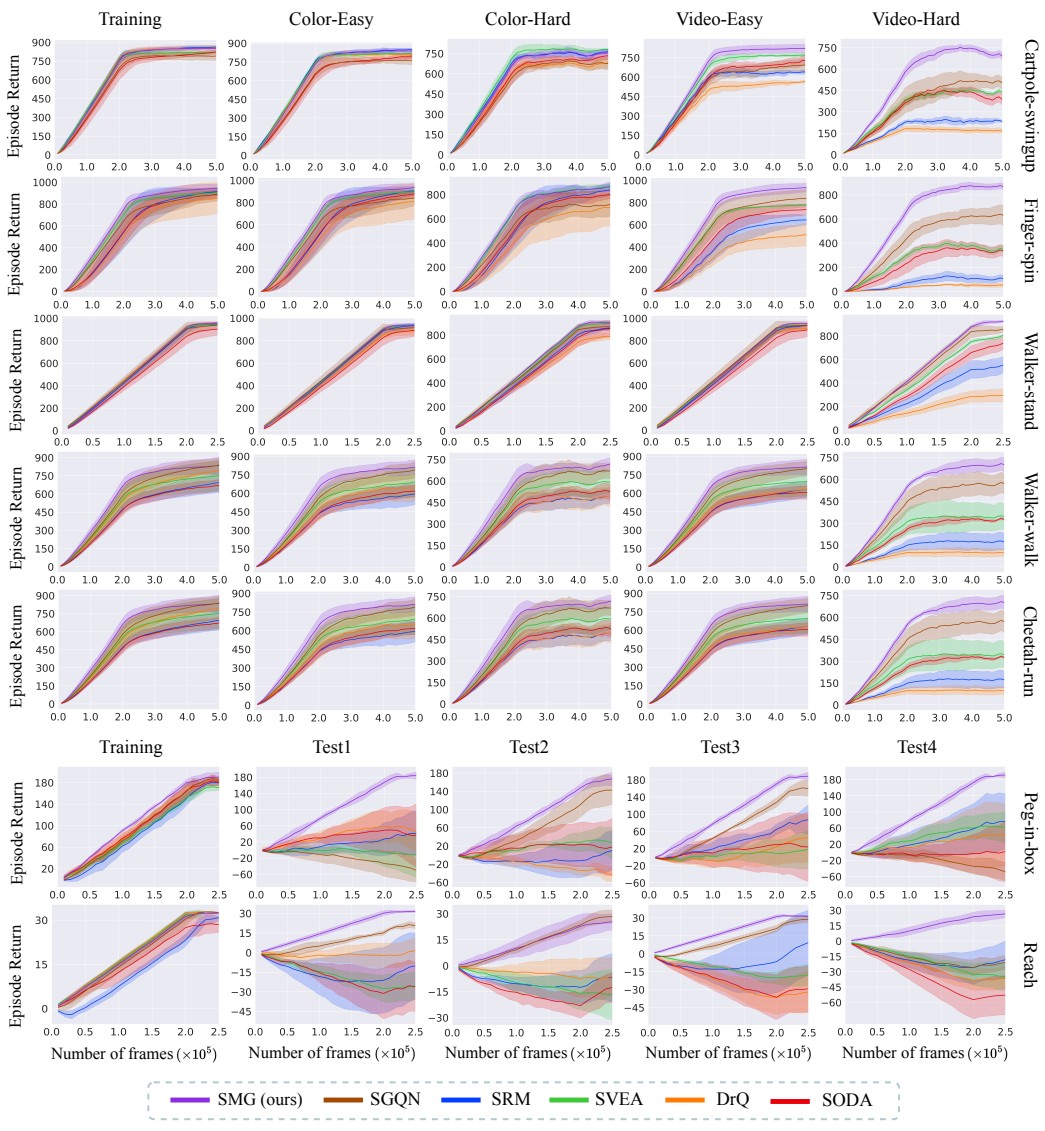

Figure 11: Training curves in all seven tasks. We evaluate each seed three times and then calculate the mean episode return for every 10k training steps, and the variance is shown as the shaded area by calculating four random seeds.

