# OpenReview forum: "Focus On What Matters: Separated Models For Visual-Based RL Generalization"
_NeurIPS.cc/2024/Conference — NeurIPS 2024 poster_

### Official Review · Reviewer_iwu6 · 2024-06-13

**Soundness:** 3
**Presentation:** 2
**Contribution:** 2
**Rating:** 5
**Confidence:** 4

**Summary:**

Visual-based Reinforcement Learning (RL) often fails to generalize across unseen environments. This work proposes SMG (Separated Models for Generalization) to improve the generalization in VRL by introducing two models to separately extract task-relevant and task-irrelevant representations through image reconstruction. Specifically, SMG proposes two additional consistency losses on relevant features, improving generalization. Extensive experiments, including video-hard DMC, color-hard DMC, and manipulation tasks, show SMG excels in diverse settings and tasks, demonstrating robust performance.

**Strengths:**

- Separating foreground and background for reconstruction makes sense for improving the generalization in VRL.

- Extensive experiments in various experimental settings demonstrate the effectiveness of SMG.

- The learned mask looks very effective (Fig. 3 and Fig. 7).

**Weaknesses:**

- Distinguishing between controllable and uncontrollable parts for learning a mask model has been widely discussed in the community, like TIA [1], Denoised MDP [2], ISO-Dream [3] and so on. Although I appreciate authors' efforts to discuss its difference against TIA (appendix E.2), I think the novelty of learning mask models to distinguish noise from the environment is limited. Nevertheless, I believe that this paper has made contributions in applying mask models to the field of visual RL generalization.

- I'm curious about the performance of the proposed method in some more challenging settings, like RL-Vigen [4].

- As there are many losses, it is better to add a detailed pseudo code about how to calculate all these losses, which can make the paper more readable.

- This proposed SGM is considered to be seamlessly combined with any existing off-policy RL algorithms. As the experiments mainly consider SAC as the RL backbone, I'm curious about its performance with other methods, like DrQ or SVEA.

- The related work part only discusses observation generalization in RL and some other types of generalization also should be discussed, like dynamic generalization [5,6] and task generalization [7,8].

Overall, I lean toward boardline of this work. I will participate in subsequent discussions and would like to adjust my scores, especially for the response to my concerns about experiments.

[1] Learning Task Informed Abstractions

[2] Denoised MDPs: Learning World Models Better Than the World Itself

[3] Iso-Dream: Isolating and Leveraging Noncontrollable Visual Dynamics in World Models

[4] RL-ViGen: A Reinforcement Learning Benchmark for Visual Generalization

[5] Context-aware dynamics model for generalization in model-based reinforcement learning

[6] Why generalization in rl is difficult: Epistemic pomdps and implicit partial observability

[7] Zero-shot task generalization with multi-task deep reinforcement learning

[8] Task Aware Dreamer for Task Generalization in Reinforcement Learning

**Questions:**

- How do you determine the hyperparameter $\rho$? In Fig.9, this paper shows different results about $\rho$ in walker-walk. Are there more results to show the relationship between SGM and $\rho$?

- Why do reconstruction-based methods benefit generalization? Are there any explanations?


---------------

**After reading the authors' response and other reviewers' comments, I have raised my scores from 4 to 5.**

**Limitations:**

Yes, this work has discussed its limitations in Sec. 6.

---

> ### Author Rebuttal · Authors · 2024-08-05
>
> We sincerely thank the reviewer for your constructive comments and suggestions. We address each of your comments as follows.
>
> ### Q1: I think the novelty of learning mask models to distinguish noise from the environment is limited.
> ---
> A1:
> Thank you for your professional analysis. However, we would like to elaborate further on the novelty of our method:
> - **The key idea behind SMG is to focus on the task-relevant features across training and testing scenarios.** The separated models structure is employed as an effective tool for extracting these task-relevant representations from observations.
> - Compared to previous approaches, SMG introduces several key innovations to enhance performance in generalization settings. **We are the first to extract reward-relevant features via the Q-network in a model-free setting, provide the background encoder with accurate supervision, use the learned foreground to bootstrap the training process, and more fully utilize the learned mask through attribution augmentation.** We believe these improvements can inspire future research to advance the separated models structure into broader fields.
> - We have added a experiment to test TIA[5] in generalization setting; please refer to the __Author Rebuttal 2.1__. As shown in Table 1, TIA fails to generalize in video-background settings in all seven tasks. This further demonstrates the effectiveness of our improvements (two consistency loss and data augmentation techniques) to the separated models architecture under the generalization settings.
>
>
> ### Q2: I'm curious about the performance of the proposed method in some more challenging settings, like RL-Vigen.
> ---
> A2:
> We have attempted to conduct experiments on three Adroit tasks introduced by [RL-Vigen](https://github.com/gemcollector/RL-ViGen/tree/ViGen-adroit?tab=readme-ov-file)[3]. However, we found that SAC-based algorithms (including SVEA, SGQN, and our SMG) perform poorly in these challenging tasks. The results reported in RL-ViGen are based on using VRL3 [4] as the backbone.
>
> VRL3 relies Adroit offline data to pre-train the encoder. When we attempted to build SMG upon VRL3, we discovered that the [download link](https://drive.google.com/drive/folders/14rH_QyigJLDWsacQsrSNV7b0PjXOGWwD?usp=sharing) for the offline data is currently invalid. As a result, we are unable to conduct experiments at this time. We have emailed the author to report this issue and will proceed with the experiments once we receive a response.
>
> Nevertheless, we report the reconstruction process of SMG in Adroit-Pen; please refer to __Author Rebuttal 2.5__. SMG successfully outputs an accurate foreground image in this challenging task, and we believe VRL3 would benefit from the learned task-relevant representation.
>
> ### Q3: As there are many losses, it is better to add a detailed pseudo code about how to calculate all these losses, which can make the paper more readable.
> ---
> A3:
> Thank you for your advice, the pseudo code is showed in __Author Rebuttal 1.4__
>
>
> ### Q4: I'm curious about its performance with other methods.
> ---
> A4:
> This is a good suggestion. We have added experiments that use SVEA[2] as a backbone; please refer to __Author Rebuttal 2.3__. The performance of SMG improves in most tasks, which is reasonable since SVEA is a stronger algorithm than SAC.
>
>
> ### Q5: Some other types of generalization also should be discussed.
> ---
> A5:
> Thank you for your reminder. We have revised Section 5; please refer to __Author Rebuttal 1.3__.
>
>
> ### Q6: How do you determine the hyperparameter $\rho$ ?
> ---
> A6:
> We invite the reviewer to refer to __Author Rebuttal 2.6__.
>
> ### Q7: Why do reconstruction-based methods benefit generalization?
> ---
> A7:
> There might be some misunderstanding. **We do not think that directly using a reconstruction-based loss benefits generalization; rather, it may exacerbate the overfitting problem to task-irrelevant features. The advantage of reconstruction-based methods lies in their ability to improve sampling efficiency [1].** Our key idea is to enhance the agent’s generalization ability while utilizing the reconstruction-based loss to improve sampling efficiency. To achieve this, we introduce the separated models (please refer to lines 39–43 of our paper).
>
> ### Reference
> ---
> [1] Improving sample efficiency in model-free reinforcement learning from images.
>
> [2] Stabilizing deep q-learning with convnets and vision transformers under data augmentation.
>
> [3] RL-ViGen: A Reinforcement Learning Benchmark for Visual Generalization.
>
> [4] Vrl3: A data-driven framework for visual deep reinforcement learning.
>
> [5] Learning task informed abstractions.

---

> > ### Comment · Reviewer_iwu6 · 2024-08-08
> > **Response to the rebuttal**
> >
> > I'd like to appreciate the authors' efforts in addressing my concerns and supplementing extra experiments.
> >
> > - Q1: Overall, I think the concept novelty of this work is limited as there are many works learning a mask model to distinguish between controllable and uncontrollable parts for handling distracting environments. However, I still believe this work has enough technique novelty to apply this insight to improving the generalization in visual RL, which is also beneficial for the community by extending relatively mature technologies to new settings.
> >
> > - Q2: Thanks for the reconstruction results in Adroit. In RL-Vigen, several settings can be handled by SAC-based algorithms like SGQN. For example, in Table-top Manipulation, RL-Vigen also considers visual appearances and lighting changes, and SGQN can handle this setting to some degree.
> >
> > - Q3-6: Thanks for the response and supplemented results, which should be added to the paper for better highlighting the paper's contribution.
> >
> > - Q7: Thanks for your clarification. There is still a small concern. As the authors have mentioned, reconstruction-based losses may exacerbate the overfitting problem but it can improve sampling efficiency. So is it more advantageous or disadvantageous for this task?
> >
> > - Other comments: I'm curious about whether this method can be extended to more general visual generalization tasks, like foreground noises, camera views changing, and so on.
> >
> > Overall, most of my concerns are addressed. Although there are several minor points, I have raised my scores into 5.

---

> > > ### Author Response · Authors · 2024-08-09
> > > **Discussion reply to the reviewer iwu6**
> > >
> > > Thank you for helping us improve the paper and update the score. We really appreciate your valuable comments!
> > >
> > > We will address your concerns as follows:
> > >
> > > - Q1: Thank you for recognizing our work! We also hope our paper can promote more research in visual-based generalization problems.
> > >
> > > - Q2: Thanks for your suggestion, we’d like to add the experiments in the camera-ready version.
> > >
> > > - Q3-6: Yes, we will revise some paragraphs and add the additional experiments to the appendix.
> > >
> > > - Q7: Overfitting to task-irrelevant features typically occurs when a single autoencoder is used to reconstruct the entire observation, as the encoder must learn all the features present in the observation. However, since SMG employs two autoencoders to separately fit task-relevant and task-irrelevant features, our model avoids such overfitting problem and can also take the advantage of high sample efficiency. But for typical methods using a single autoencoder, it's definitely a disadvantage in generalization tasks.
> > >
> > > - Other comments: We'd like to discuss these settings separately：
> > >   -  Foreground noise: SMG is particularly suited for such setting, as we use the extracted foreground for data augmentation. With a small modification, such as adding some noises to the foreground image (e.g., color jitter) before synthesizing it with the background image during the data augmentation stage, we can successfully simulate the testing scenario during training.
> > >   - Camera views changing: This setting is relatively more challenging due to the involvement of spatial transformations. Directly applying SMG to this setting may not work, as SMG is trained with a fixed camera view. However, this issue could be addressed by incorporating an additional camera during training and adding STN blocks[1] to our model, adopting a similar approach to MoVie[2].
> > >   - Overall, when applying SMG to different generalization settings, we should first consider the similarity between the data augmentation types and the testing scenarios, and then adjust the augmentation methods accordingly.
> > >
> > >
> > > ---
> > > __Reference__
> > >
> > > [1] Spatial transformer networks.
> > >
> > > [2] Movie: Visual model-based policy adaptation for view generalization.

---

> > > > ### Comment · Reviewer_iwu6 · 2024-08-09
> > > > **Thanks for your response**
> > > >
> > > > I'd like to thank the authors for the detailed and thorough response. I remain positive about this paper and believe it will be of interest to researchers of generalization in visual RL.

---

### Official Review · Reviewer_ZRY4 · 2024-07-08

**Soundness:** 3
**Presentation:** 3
**Contribution:** 3
**Rating:** 7
**Confidence:** 4

**Summary:**

This paper presents a novel method that utilizes two model branches to extract task-relevant and task-irrelevant representations separately from visual observations, aiming to enhance the zero-shot generalization ability of RL agents. The approach introduces four additional loss terms and two consistency losses to guide the agent's focus towards task-relevant areas across different scenarios. The proposed method can be seamlessly integrated into existing standard off-policy RL algorithms as a plug-and-play module. Experimental results demonstrate the effectiveness of the proposed model on two environments, surpassing previous benchmarks such as SAC and DrQ.

**Strengths:**

1. This paper is clearly written and easy to follow.
2. Based on the separated models architecture, this paper proposes multiple effective loss functions to focus on task-relevant features in visual-based RL generalization.
3. The authors provide detailed validations on the DMC environment and robotic manipulation tasks. They demonstrate the advantages of the proposed loss terms across multiple tasks in DMC (Table 3) and showcase the state-of-the-art performance of SMG (Table 1, 2).

**Weaknesses:**

1. While the paper compares the performance with model-free RL methods, it would be beneficial to also include a comparison with model-based RL methods. Previous works such as DreamerPro [1], Iso-Dream [2], and Denoised-MDP [3] have addressed visual distractions to enhance the generalization ability of RL agents.

[1] Dreamerpro: Reconstruction-free model-based reinforcement learning with prototypical representations.

[2] Iso-Dream: Isolating Noncontrollable Visual Dynamics in World Models.

[3] Denoised mdps: Learning world models better than the world itself.

2. The paper lacks sufficient discussion and analysis of its limitations.
3. The serial numbers in some figures appear to be somewhat disorganized.

**Questions:**

Please see the weaknesses section.

**Limitations:**

This paper discusses the limitations, but that is not enough.

---

> ### Author Rebuttal · Authors · 2024-08-05
>
> We sincerely thank the reviewer for your constructive comments and suggestions. We address each of your comments as follows.
>
> ### Q1: It would be beneficial to also include a comparison with model-based RL methods.
> ---
> A1:
> To the best of our knowledge, there are still no model-based methods that effectively address the generalization problem in visual-based RL. Therefore, in our original paper, we did not include comparisons with model-based methods.
>
> Although a series of model-based methods (e.g., TIA[1], DenoisedMDPs[2]) use a similar separated models structure like SMG, they address a fundamentally different problem. These algorithms primarily aims to enhance the agent’s robustness to distractors during training. Specifically, TIA trains and tests both in video background environments, whereas our method trains without background and tests in the video background environments (just like the difference between I.I.D and OOD). Methods like TIA do not incorporate specific designs, such as data augmentation techniques, to bridge the gap between training and testing scenarios, and therefore lack generalization capabilities.
>
> Nevertheless, we have included TIA as a baseline to support our point; please refer to the __Author Rebuttal 2.1__. As shown in table 1, TIA fails to generalize in video-background settings in all seven tasks. This further demonstrates the effectiveness of our improvements (two consistency loss and data augmentation techniques) to the separated models architecture under the generalization settings.
>
> ### Q2: The paper lacks sufficient discussion and analysis of its limitations.
> ---
> A2:
> Thank you very much for the reminder, we have added a paragraph to Section 6 that describes the limitation of our work. Please refer to the __Author Rebuttal 1.2__.
>
> ### Q3: The serial numbers in some figures appear to be somewhat disorganized.
> ---
> A3: Thank you for your feedback. The order of Figures 3 and 4 is indeed incorrect due to LaTeX’s automatic formatting. We will ensure this is corrected in the final version of the paper.
>
> ### Reference
> ---
>
> [1] Learning task informed abstractions.
>
> [2] Denoised mdps: Learning world models better than the world itself.

---

### Official Review · Reviewer_jV8P · 2024-07-12

**Soundness:** 3
**Presentation:** 3
**Contribution:** 2
**Rating:** 5
**Confidence:** 4

**Summary:**

This paper presents a novel approach called SMG (Separated Models for Generalization) to improve generalization in visual-based reinforcement learning (RL). The approach works by using separate foreground and background encoders/decoders and employing a mask to isolate task-relevant regions. In addition, it also applies four additional losses(mask ratio, background, Q-value and empowerment losses) to to enhance the model’s ability to distinguish between two types of representations. To make the learned models generalize to different visual styles, it introduces attribution augmentation and consistency losses. The authors position this as a plug-and-play method that can enhance existing RL algorithms' generalization capabilities.

Experiments show SMG outperforms baseline methods, particularly in video-background settings where it maintains performance even with significant visual changes. Ablation studies validate the importance of each component.

The main contributions are:

- SMG: A separated model architecture with two branches to extract task-relevant and task-irrelevant representations from visual observations.

- Two consistency losses to guide the agent's focus on task-relevant areas across different scenarios.

- Strong performance on DMControl benchmark tasks, especially in video-background settings.

**Strengths:**

This paper has several strengths:

- SMG achieves state-of-the-art performance on the DMControl Generalization Benchmark, particularly excelling in the challenging video-background settings. This demonstrates the practical effectiveness of the approach.

- SMG is a plug-and-play method that can enhance existing RL algorithms' generalization capabilities. It is designed to be easily integrated with existing off-policy RL algorithms, enhancing its practical value and potential for wide adoption.

- This paper includes detailed ablation studies that validate the importance of each component in the SMG architecture, providing insights into the method's workings.

- This paper is well-written. And it also provides clear visualizations of the reconstruction process, helping readers understand how SMG extracts and utilizes task-relevant features.

**Weaknesses:**

This paper has several weaknesses:

- My major concern is the overclaim made by this paper. While it claims to address the generalization gap in visual-based reinforcement learning, the method proposed primarily tackles scenarios where only the backgrounds differ. However, visual generalization challenges are more diverse and include variations such as different lighting conditions and textures, which are common in real-world robotics applications. These scenarios appear to be overlooked in this paper.

- SMG introduces a lot of loss terms and associated hyperparameters, which could complicate tuning in practical applications.
    - Specifically, the mask ratio $\rho$ appears to be crucial for performance, as it is the sole factor preventing the model from classifying everything as foreground. Given that $\rho$ represents the ratio between the foreground and the entire image, it likely necessitates per-task tuning, which could prove to be challenging and not scalable.

- The foreground consistency loss, as discussed in Section 3.3, heavily depends on the predicted mask to construct the augmented observation. During the initial stages of training, this process relies on potentially inaccurate mask predictions and attributions. Although the authors describe this as a bootstrapping process, further analysis regarding its stability and potential failure modes would be beneficial.

- The paper could be strengthened by considering a broader range of baselines. For example:
    - Recent studies [1] suggest that visual encoders pre-trained on large-scale image datasets can improve the visual robustness of a policy. This paper does not make any comparisons with visual pre-training methods.
    - Large vision foundation models like SAM [2] could potentially be utilized to provide supervision for generating foreground masks. Would this approach be more effective than training a mask predictor from scratch?


- The additional computation overhead introduced by the extra modules is concerning.
    - The architecture, which involves separate models, essentially doubles the number of parameters compared to baseline methods. Although the authors argue that the performance improvements are due to the novel architecture rather than the increased number of parameters, this could still be problematic for practical applications with limited computational resources.
    - Training time: The reported wall time for SMG is significantly longer than that of the baseline methods (22 hours versus 8-13 hours for 500,000 steps).

[1] Hansen, Nicklas, et al. "On pre-training for visuo-motor control: Revisiting a learning-from-scratch baseline." arXiv preprint arXiv:2212.05749 (2022).

[2] Kirillov, Alexander, et al. "Segment anything." Proceedings of the IEEE/CVF International Conference on Computer Vision. 2023.

**Questions:**

See the "Weaknesses" section. Some additional questions are noted below:

- The terminology used for "foreground" and "background" is somewhat confusing. To clarify, "foreground" actually refers to the task-relevant parts of the image, while "background" refers to the task-irrelevant parts, correct?
- The necessity for background reconstruction is unclear. The authors claim that "improving the precision of background prediction can consequently enhance the foreground as well," but a more detailed explanation of this assertion would be beneficial.

**Limitations:**

There is no separate section for limitations in the paper.

---

> ### Author Rebuttal · Authors · 2024-08-05
>
> We sincerely thank the reviewer for your constructive comments and suggestions. We address each of your comments as follows.
>
> ### Q1: Some testing scenarios appear to be overlooked in this paper.
> ---
> A1:
> We admit that real-world deployment scenarios can be more diverse and complex. **However, the four testing scenarios used in our paper are widely accepted benchmarks in visual-based RL [3,4,5].** For example, the color variation settings simulate different illumination conditions, while the video variation settings simulate training a robot in different environments, such as training indoors and testing outdoors. In addition, the testing scenarios in the two robotic manipulation tasks simulate texture changes, Therefore, we don’t think the real scenarios are overlooked.
>
> Moreover, no algorithm has yet effectively solved the video-hard setting except ours, which is the closest scenario to realistic applications. We hope our paper can promote more research in this challenging setting.
>
>
> ### Q2: SMG introduces a lot of loss terms.
> ---
> A2:
> This phenomenon is common in methods that utilize separate models. For example, DenoisedMDPs [3] has four loss terms, Iso-dreamer [4] has six loss terms, TIA [2] has seven loss terms, and IFactor [5] has eight loss terms, yet all achieve excellent performance. Therefore, we believe the focus should be on the difficulty of adjusting the loss weights for different tasks rather than on the number of loss terms.
>
> **In all seven tasks, we used the same weights for the five auxiliary loss terms.** The results demonstrate that weighting four of the five terms equally is sufficient to achieve outstanding performance. This indicates that our method is robust enough across different tasks and does not require additional time for adjusting the weights.
>
> ### Q3: About mask ratio $\rho$
> ---
> A3:
> Please refer to __Author Rebuttal 2.6__
>
> ### Q4: Although the authors describe this as a bootstrapping process, further analysis regarding its stability and potential failure modes would be beneficial.
> ---
> A4:
> Thanks for rising this concern.
> **The key guarantee that ensures the models learns accurate foreground images and masks is the utilization of $L_{q}$ and $L_{action}$ to extract reward-relevant and action-relevant features.** The bootstrapping process then starts based on these features.
>
> We have added an experiment to show a failure case by removing $L_{action}$ and stopping the gradient from $L_{q}$ to $z^+$; please refer to __Author Rebuttal 2.7__. As shown in Figure 3, the foreground models failed to learn meaningful images, and the background models learned everything from the input observations. This failure case occurs approximately once every five seeds. However, when we add the two loss terms back, we have never observed such failures.
>
>
> ### Q5: The paper could be strengthened by considering a broader range of baselines.
> ---
> A5:
> Thanks for your suggestion. We have added two more baselines:
> - SAM-G [3] is a method that utilizes both pre-trained image encoders and the Segment Anything Model. **Although the results reported in the paper use four times the batch size and twice the training steps compared to SMG, SMG still achieves comparable performance.** This is because methods with pre-trained models heavily rely on the similarity between pre-training datasets and the RL task scenarios. Utilizing Large vision foundation models also increasing parameters and training time (SAM-G takes around 2d to train). Despite being trained from scratch, SMG can quickly distinguish foreground and background parts from environments and is more flexible than these pre-trained methods. Please refer to __Author Rebuttal 2.2__.
>
> - TIA[1] is a model-based method that uses a similar separated models methods to us. However, as TIA is not designed for visual generalization, it fails in our testing scenarios. Please refer to __Author Rebuttal 2.1__.
>
>
> ### Q6: The additional computation overhead introduced by the extra modules is concerning.
> ---
> A6:
> We agree that algorithms are required to be time efficient to train in realistic tasks, but we do think the time consumption of SMG is reasonable.
>
> - Although SMG requires 22 hours to train for 500k time steps, **in almost all tasks, SMG already outperforms the baselines’ 500k performance at 200k time steps (please refer to Figure 11 in our original paper).** We trained for 500k time steps to ensure a fair comparison. Therefore, it is possible to train for only half the time reported in our paper.
>
> - Algorithms that perform well in more complex tasks inevitably lead to more complicated model structures and an increase in the number of parameters. Despite the fact that baselines training consume less time, they simply can't generalize at all in the robotic manipulation tasks. And we think the training time for SMG is acceptable as methods like SAM-G cost much more training time.
>
>
> ### Q7: "foreground" actually refers to the task-relevant parts of the image, while "background" refers to the task-irrelevant parts, correct?
> ---
> A7:
> Yes, you are right. We think the terminologies “foreground” and “background” are more accessible for readers to understand our key idea. Therefore, we will add a clear definition in Section 3.1:
>
> “In the following paragraphs, we will use ‘foreground’ to refer to task-relevant parts and ‘background’ to refer to task-irrelevant parts.”
>
> ### Q8: The necessity for background reconstruction is unclear.
> ---
> A8:
> Please refer to __Author Rebuttal 1.5__
>
> ### Q9: There is no separate section for limitations in the paper.
> ---
> A9:
> Please refer to __Author Rebuttal 1.2__
>
> ### Reference
> ---
> [1] Learning task informed abstractions.
>
> [2] Learning world models with identifiable factorization.
>
> [3] Generalizable visual reinforcement learning with segment anything model.
>
> [4] Stabilizing deep q-learning with convnets and vision transformers under data augmentation.
>
> [5] Generalization in Reinforcement Learning by Soft Data Augmentation

---

> > ### Comment · Reviewer_jV8P · 2024-08-10
> >
> > Thank you to the authors for their response.
> >
> > I still feel that the proposed method appears to be tailored primarily for generalization across varying visual backgrounds. I would recommend that the authors consider moderating the scope of their claims accordingly.

---

> > > ### Author Response · Authors · 2024-08-11
> > > **Discussion reply to reviewer jV8P**
> > >
> > > Thanks for your response. There seems to be some misunderstanding regarding our method and the experimental settings. SMG is definitely not tailored specifically for visual background settings, and we’d like to elaborate further:
> > >
> > > - SMG is not primarily tackles scenarios where only the backgrounds differ. The key idea behind SMG is to focus on the task-relevant features across training and testing scenarios, such idea is applicable to any generalization tasks. Although more diverse generalization scenarios exist in the real world, we have tried to include a sufficient and widely accepted range of test settings in the paper to demonstrate the generalization ability of SMG.
> > >
> > > - We adopt a widely used evaluation setting similar to previous works [1,2,3,4], which includes color-easy, color-hard, video-easy, video-hard, and five different testing scenarios in robotic manipulation tasks. Only two of the nine settings are related to visual backgrounds: the other two color settings in DMC introduce randomized colors (both the foreground and background), while the five settings in robotic manipulation tasks involve changes to both color and textures (please refer to figure 5 and figure 6 in our paper).
> > >
> > > - Despite SMG’s impressive performance in the video background settings, we’d like to highlight the improvements SMG delivers in other settings as well. In the random-color settings (Table 6 in our paper), SMG outperforms all baselines in 7 out of 10 tasks, with the performance gap within 5% in the other 3 tasks. In robotic manipulation tasks (Table 2 in our paper), SMG is the only method to maintain stable performance across 5 random-texture settings. These results indicate that SMG not only performs well in video-background settings but also exhibits superior generalization capability in random-color and random-texture settings.
> > >
> > >
> > > ---
> > >
> > > Other than that, I hope the answers to your other questions have addressed your queries. If there are any additional concerns, please let us know!
> > >
> > > ---
> > > Reference
> > >
> > > [1] Generalization in Reinforcement Learning by Soft Data Augmentation.
> > >
> > > [2] Stabilizing Deep Q-Learning with ConvNets and Vision Transformers under Data Augmentation.
> > >
> > > [3] Look where you look! Saliency-guided Q-networks for generalization in visual Reinforcement Learning.
> > >
> > > [4] Spectrum Random Masking for Generalization in Image-based Reinforcement Learning.

---

> > > > ### Comment · Reviewer_jV8P · 2024-08-11
> > > >
> > > > Thank you for the explanation. I have updated my rating.

---

> > > > > ### Author Response · Authors · 2024-08-11
> > > > > **Discussion reply to reviewer jV8P**
> > > > >
> > > > > Thank you for helping us improve the paper and update the score, we really appreciate your valuable comments!

---

### Official Review · Reviewer_C7ia · 2024-07-12

**Soundness:** 3
**Presentation:** 2
**Contribution:** 2
**Rating:** 7
**Confidence:** 3

**Summary:**

The authors propose a novel objective to improve robustness of the visual encoder in RL to background noise and to color perturbations. First, the authors split the visual encoder into two models: background encoder/decoder and foreground encoder/decoder. The proposed training objective contains multiple components:
- overall reconstruction loss that combines outputs of the background and the foreground decoders modulated by a mask;
- mask ratio loss that prevents the foreground mask from taking up too much of the image;
- background reconstruction loss that uses the learned mask to generate a new data sample;
- q-value loss that makes the foreground representation capture value information;
- empowerment loss that makes the foreground representations capture the agent actions;
- foreground and q-value consistency losses that make sure that changing the background (using the learned mask) doesn't change the foreground features and q-values

The method is tested on DMC generalization benchmark and on robotic manipulation tasks.

**Strengths:**

- The method performs really well with various distractors;
- The idea of re-using the learned masks for augmentations is interesting and, as far as I can tell, novel;

**Weaknesses:**

- The writing is a bit sloppy, with many typos and confusing sentences;
- The resulting objective is too complex and has too many terms;
- No comparison to TIA, although the presented method is quite similar. Was that because you only compare to model-free methods?

Typos (some of them, I didn't write down all of them, please run a spell checker on the text):
line 13: achieving free from overfitting : not clear what this means
line 38: further strengths -> further strengthens
line 100: focused in -> focused on
line 536: Comparision -> comparison

**Questions:**

- In your objectives, you're maximizing the mutual information between foreground representations of two consecutive states and the action that was taken between them. Have you tried minimizing MI between background representations and actions and or rewards? This could be done with information bottleneck method for example. If you haven't tried this, do you this can help?

**Limitations:**

The authors have adequately described limitations and potential negative societal impact of their work.

---

> ### Author Rebuttal · Authors · 2024-08-04
>
> We sincerely thank the reviewer for your constructive comments and suggestions. We address each of your comments as follows.
>
> ### Q1: The writing is a bit sloppy, with many typos and confusing sentences
> ---
> A1:
> Thank you for your careful review. We have thoroughly reviewed the paper multiple times and assure you that the errors listed in __Author Rebuttal 1.1__ will be corrected in the final version of the paper.
>
> We hope that the current version is clear enough for readers to understand our key ideas. If there are any sentences that are still prone to misunderstanding, please let us know.
>
> ### Q2: The resulting objective is too complex and has too many terms
> ---
> A2:
> This phenomenon is common in methods that utilize separate models. For example, Iso-dreamer [4] has six loss terms, TIA [2] has seven loss terms, and IFactor [5] has eight loss terms, yet all achieve excellent performance. Therefore, we believe the focus should be on the difficulty of adjusting the loss weights for different tasks rather than on the number of loss terms.
>
> **In all seven tasks, we used the same weights for the five auxiliary loss terms.** The results demonstrate that weighting four of the five terms equally is sufficient to achieve outstanding performance. This indicates that our method is robust enough across different tasks and does not require additional time for adjusting the weights.
>
> The main concern may stem from why we use a smaller weight for $L_{fore\ consist}$ than others. This is because a too-large weight would lead to the model overfitting the inaccurate attribution predictions in the early stage (as we use the model output under raw observation as ground truth), and cause the foreground encoder cost more time steps to learn an accurate task-relevant representation. **We have added a new ablation study to provide readers with more insight into our loss weight setting.** By setting $\lambda_{fore\ consist}$ to 1 (refer to __Author Rebuttal 2.4__), the performance of SMG drops around 10~20% across different tasks.
>
> ### Q3: No comparison to TIA, although the presented method is quite similar. Was that because you only compare to model-free methods?
> ---
> A3:
> To the best of our knowledge, there are still no model-based methods that effectively address the generalization problem in visual-based RL. Therefore, in our original paper, we did not include comparisons with model-based methods.
>
> Although a series of model-based methods (e.g., TIA[2], DenoisedMDPs[3]) use a similar separated models structure like SMG, they address a fundamentally different problem. **These algorithms primarily aims to enhance the agent’s robustness to distractors during training.** Specifically, TIA trains and tests both in video background environments, **whereas our method trains without a background and tests in the video background environments (just like the difference between I.I.D and OOD).** Methods like TIA do not incorporate specific designs, such as data augmentation techniques, to bridge the gap between training and testing scenarios, and therefore lack generalization capabilities.
>
> Nevertheless, we have included TIA as a baseline to support our point; please refer to the __Author Rebuttal 2.1__. As shown in Table 1, TIA fails to generalize in video-background settings in all seven tasks. This further demonstrates the effectiveness of our improvements (two consistency loss and data augmentation techniques) to the separated models architecture under the generalization settings.
>
>
> ### Q4: Have you tried minimizing MI between background representations and actions and or rewards?
> ---
> A4:
> This is a very interesting question. Why we aim to maximize MI between foreground representation and actions or rewards is because this is the most straightforward way to learn a good task-relevant representation. We did consider minimizing the MI between background representation and actions or rewards as well, but we abandoned this for the following reasons:
>
> TIA[2] attempts to minimize the mutual information between the background representation and rewards with the loss term $L_{Radv}=-max_qq(r_t|s_t^-)$ . TIA implements this by maximizing the prediction loss of the background reward model (https://github.com/kyonofx/tia/blob/main/Dreamer/dreamers.py#L368). This approach appears problematic because the reward model can easily learn parameters that satisfy this condition, such as outputting 0 for any state, which would make the prediction loss consistently high. We have also done ablation experiments on TIA before, and the results show that this loss term does not improve the model performance significantly. Given that incorporating this term would further complicate the optimization objective, we chose not to include it.
>
> As for the MI between the background representation and actions, since the background encoder does not include actions as its input, the background representation inherently lacks action-relevant features. Therefore, there is no need to minimize this MI term.
>
>
>
> ### Reference
> ---
> [1] Generalizable visual reinforcement learning with segment anything model.
>
> [2] Learning task informed abstractions.
>
> [3] Denoised mdps: Learning world models better than the world itself.
>
> [4] Iso-dream: Isolating and leveraging noncontrollable visual dynamics in world models.
>
> [5] Learning world models with identifiable factorization.

---

> > ### Comment · Reviewer_C7ia · 2024-08-13
> >
> > Thank you for clarifying the difference with TIA and running the additional experiments.
> > Also thank you for clarifying that you use the same hyperparameters across all tasks. Please emphasize that in the text, I think that strengthens your claims.
> >
> > I increase my score to 7.

---

> > > ### Author Response · Authors · 2024-08-14
> > > **Discussion reply to reviewer C7ia**
> > >
> > > Yes, we will revise some paragraphs and add the additional experiments to the appendix. Thank you again for helping us improve the paper and update the score, we really appreciate your valuable comments!

---

### Author Rebuttal · Authors · 2024-08-05

We revised the paper and added suggested experiments according to the reviewer’s comments. The detailed revisions are described as follows. The additional figures and a table are attached in the pdf file.

# 1. Revisions

### 1.1. Revise some typos and sentence

line 13: achieving free from overfitting -> achieving free from overfitting to task-irrelevant features.

line 38: further strengths -> further strengthens

line 100: focused in -> focused on

line 458: perceptrons -> perceptions

line 523: Bootstraping -> Bootstrapping

line 536: Comparision -> comparison

### 1.2. Added a paragraph describing limitations in section 6

SMG is particularly well-suited for robotic manipulation tasks in realistic scenarios. However, when the observation contains too many task-relevant objects, the complexity of accurately learning a mask increases. This can lead to a decline in SMG’s performance. For instance, in an autonomous navigation task, the presence of numerous pedestrians in the view makes it challenging to accurately mask all of them.

### 1.3. Introduce more related works in section 5

In addition to view generalization, considerable research has focused on dynamic generalization [3,4,5] to develop a global model capable of generalizing across different dynamics. Additionally, several studies [6,7,8] have explored task generalization, which aims to enable learned agents to generalize to new tasks.

### 1.4. Add the pseudo code of SMG

| __Algorithm 1__ SAC with separated models |
| :----|
|__Denote__ network parameters $\theta$, mask ratio ρ, batch size $N$, replay buffer $\mathcal{B}$|
|__Denote__ policy network $\pi_{\theta}$, foreground encoder $f^+_{\theta}$,background encoder $f^-\_{\theta}$|
|__for__ each iteration time step __do__|
|$\qquad a,o',r\sim\pi_{\theta}(f^+_{\theta}(o)),\mathcal{P}(o,a),\mathcal{R}(o,a)$|
|$\qquad \mathcal{B}\leftarrow \mathcal{B}\ ∪\ (o,a,r,o')$|
|$\qquad$ __for__ each update time step __do__|
|$\qquad \qquad \\{o_i,a_i,r_i,o'\_i\\}_{i\in[1,N]}\sim \mathcal{B}$|
|$\qquad \qquad o^+\_i,mask_i\sim f^+_{\theta}(o_i)$|
|$\qquad \qquad o^-\_i\sim f^-_\theta(o_i)$|
|$\qquad \qquad o^{aug}_i \leftarrow o^+\_i\ast mask_i+\epsilon\ast(1-mask_i)$ // $\epsilon$ is sampled from image dataset|
|$\qquad \qquad L_{recon}\leftarrow L(o_i,\ o^+_i\ast mask_i+o^-_i\ast(1-mask_i))$ // Equation 2|
|$\qquad \qquad L_{fore\ consist}\leftarrow L(o^+\_i,f^+_\theta(o^{aug}_i))$ // Equation 7|
|$\qquad \qquad L_{back}\leftarrow L(\epsilon,f^-_\theta(o^{aug}_i))$ // Equation 4|
|$\qquad \qquad L_{action}\leftarrow L(o_i,o'_i,a)$ // Equation 6|
|$\qquad \qquad L_{mask}\leftarrow L(mask_i,\rho)$ // Equation 3|
|$\qquad \qquad L_{q\ consist}\leftarrow L(Q_\theta(f^+\_\theta(o_i),a),Q\_\theta(f^+_\theta(o^{aug}_i),a))$ // Equation 8|
|$\qquad \qquad L_{aux}\leftarrow L_{recon}+L_{fore\ consist}+L_{back}+L_{action}+L_{mask}$ // auxiliary loss|
|$\qquad \qquad L_{critic}\leftarrow L_{q}+L_{q\ consist}$ // critic loss|
|$\qquad \qquad$ __update__ $\theta$ with $L_{actor},L_{critic},L_{aux}$|
|$\qquad$ __end for__|
|__end for__|
|$L_{q},L_{actor}$ is defined by SAC|
||

### 1.5 Revise line 136

"Improving the precision of background prediction can consequently enhance the foreground as well. Since the foreground and background are complementary, providing supervision for the background prevents the foreground from learning all parts of the observation."

# 2. Experiments

### 2.1. Add TIA [2] as a new baseline

The results are shown in the 4th column of __Table 1__.

### 2.2 Add SAM-G [1] as a new baseline

The results are shown in the 5th column of __Table 1__.

Training SAM-G for one seed on our RTX3090 graphics card took nearly two days. Therefore, we are currently unable to complete all the training during the rebuttal period. Instead, we report the results from the SAM-G paper, noting that they used four times the batch size and twice the training time steps compared to SMG. Despite this, SMG still achieves comparable performance to SAM-G.

### 2.3. Add SVEA as a new backbone method

The results are shown in the 8th column of __Table 1__.

### 2.4. Add a new ablation of $L_{fore}$

The results are shown in the 6th column of __Table 1__.


### 2.5. Visualize the reconstruction process in adroit-pen

The results are shown in the __Figure 1__.

### 2.6. Further study the impact of ρ in peg-in-box

ρ does need to be set for different tasks but the choice of ρ can be roughly estimated based on the percentage of the task-relevant area. It is also possible to directly reuse settings from similar tasks. For instance, we used ρ=0.06 for three different tasks and ρ=0.12 for two different tasks.

In addition, an imprecise ρ only slightly affects the performance. We have added a new ablation for ρ in peg-in-box task with a much larger interval; The results in Figure 2 indicate that variations do not significantly influence the performance. As the optimal ρ being 0.12, even when setting ρ to 0, SMG still accurately masks out the task-relevant area and the performance drops by only 8%.

### 2.7. Add a new ablation of removing $L_{q}$ and $L_{action}$

The results are shown in the __Figure 3__.


# Reference

[1] Generalizable visual reinforcement learning with segment anything model.

[2] Learning task informed abstractions.

[3] Context-aware dynamics model for generalization in model-based reinforcement learning.

[4] Why generalization in rl is difficult: Epistemic pomdps and implicit partial observability.

[5] Graph networks as learnable physics engines for inference and control.

[6] Zero-shot task generalization with multi-task deep reinforcement learning.

[7] Task Aware Dreamer for Task Generalization in Reinforcement Learning.

[8] Learning modular neural network policies for multi-task and multi-robot transfer.

---

### Decision · Program_Chairs · 2024-09-25

**Decision:**

Accept (poster)

**Comment:**

This submission proposes a new method to enhance the generalization capabilities of visual-based reinforcement learning (RL). The proposed method is composed of two branches to independently extract task-relevant and task-irrelevant features from visual inputs, utilizing a mask to isolate critical areas. It then combines multiple loss values to increase the robustness across different environments. By its simple design, it can be complementary to standard RL methods. The authors showed that the proposed method outperforms baseline methods, even on challenging cases with visual perturbations.

All authors agreed that this is a good submission to be shared with the community, and the meta reviewer also agrees with the reviewers. It is a clear accept!